# An RNAi screen unravels the complexities of Rho GTPase networks in skin morphogenesis

Melanie Laurin, Nicholas C Gomez, John Levorse, Ataman Sendoel, Megan Sribour, Elaine Fuchs*

Robin Neustein Laboratory of Mammalian Cell Biology and Development, Howard Hughes Medical Institute, The Rockefeller University, New York, United States

**Abstract** During mammalian embryogenesis, extensive cellular remodeling is needed for tissue morphogenesis. As effectors of cytoskeletal dynamics, Rho GTPases and their regulators are likely involved, but their daunting complexity has hindered progress in dissecting their functions. We overcome this hurdle by employing high throughput in utero RNAi-mediated screening to identify key Rho regulators of skin morphogenesis. Our screen unveiled hitherto unrecognized roles for Rho-mediated cytoskeletal remodeling events that impact hair follicle specification, differentiation, downgrowth and planar cell polarity. Coupling our top hit with gain/loss-of-function genetics, interactome proteomics and tissue imaging, we show that RHOU, an atypical Rho, governs the cytoskeletal-junction dynamics that establish columnar shape and planar cell polarity in epidermal progenitors. Conversely, RHOU downregulation is required to remodel to a conical cellular shape that enables hair bud invagination and downgrowth. Our findings underscore the power of coupling screens with proteomics to unravel the physiological significance of complex gene families.

DOI: https://doi.org/10.7554/eLife.50226.001

*For correspondence:
fuchslb@rockefeller.edu

## Introduction

During embryonic development, fundamental cell fate decisions are driven by a succession of reciprocal and antagonistic interactions between developmental signaling pathways that elicit changes in gene expression. Accompanied by these events are dramatic and diverse cellular rearrangements typify tissue morphogenesis (*Van Aelst and Symons, 2002*; *Gilmour et al., 2017*). Due to their ability to regulate cytoskeletal and cellular junction remodeling, Rho GTPases and their regulators are emerging as key effectors of these fascinating morphogenetic dynamics (*Van Aelst and Symons, 2002*; *Duquette and Lamarche-Vane, 2014*). Yet, a complete picture of the physiological roles played by these networks in mammals is lacking.

Most Rho GTPases act as molecular switches that remain inactive when bound to GDP, but become active when bound to GTP, where they can associate with their downstream effectors and elicit changes in cellular activity (*Vetter and Wittinghofer, 2001*; *Cherfils and Zeghouf, 2013*). Rho GTPases are regulated by Guanine nucleotide exchange factors (RhoGEFs) that promote the exchange of GDP for GTP (*Laurin and Cote, 2014*; *Cook et al., 2014*), GTPase-activating proteins (RhoGAPs) that catalyze the hydrolysis of GTP to GDP (*Olson, 2018*; *Tcherkezian and Lamarche-Vane, 2007*) and guanine nucleotide dissociation inhibitors (RhoGDI) that prevent reactivation from the GDP to GTP bound state (*Garcia-Mata et al., 2011*). Hence, fine-tuning of Rho GTPase-mediated cellular events is achieved through the protein's GTP status, localization, effector-binding, as well as gene expression and protein stability (*Hodge and Ridley, 2016*; *Olson, 2018*).

In mammals, there are 20 Rho GTPases and nearly 150 regulators. Cell culture screening approaches have uncovered roles played by Rho GTPases and their regulators in cellular processes (*Duquette and Lamarche-Vane, 2014*; *Amado-Azevedo et al., 2018*; *Tajadura-Ortega et al., 2018*; *Pascual-Vargas et al., 2017*). Thus far however, in vivo analyses of Rho GTPase functions in morphogenetic processes have been largely restricted to signaling by the three prototypical members RAC, RHO and CDC42 (*Hodge and Ridley, 2016*; *Heasman and Ridley, 2008*; *Van Aelst and Symons, 2002*; *Duquette and Lamarche-Vane, 2014*). Given the associated risks of functional redundancies, few researchers have ventured into the realm of employing conventional genetic methods to tackle the physiological roles of other family members. This has left an incomplete picture of the biological significance of the multiplicity of Rho GTPases, and the ways in which their associated actomyosin dynamics govern the complex morphogenetic events that are needed for tissues development and function.

The skin epithelium is an exceptional example of the dramatic and diverse changes in cytoskeletal-adhesion dynamics that occur during tissue development. The surface ectoderm emerges at embryonic day E9.5, shortly after gastrulation. As yet unspecified, these progenitors soon transition from a single layer of progenitors to a stratified epidermis replete with hair follicles (HFs) (*Gonzales and Fuchs, 2017*; *Jamora et al., 2003*; *Perez-Moreno and Fuchs, 2006*; *Lechler and Fuchs, 2007*; *Devenport and Fuchs, 2008*; *Devenport et al., 2011*; *Luxenburg et al., 2011*; *Luxenburg et al., 2015*; *Cohen et al., 2019*). By E12.5, epidermal progenitors begin to stratify. This process requires actomyosin dynamics and epithelial polarization, driven basally by integrin-mediated assembly and adhesion to an underlying basement membrane of extracellular matrix proteins, and laterally by formation of cadherin-mediated intercellular junctions (*Niessen et al., 2012*; *Simpson et al., 2011*; *Raghavan et al., 2003*; *Vaezi et al., 2002*; *Lechler and Fuchs, 2005*; *Perez-Moreno and Fuchs, 2006*; *Luxenburg et al., 2011*; *Luxenburg et al., 2015*). Fate specification begins around E13.5 and continues until birth in four sequential waves, as some epidermal progenitors accumulate sufficient instructive WNT-cues to coalesce and form a hair placode and an associated underlying dermal condensate (*DasGupta and Fuchs, 1999*; *Saxena et al., 2019*) (*Figure 1A*, adapted from *Devenport and Fuchs, 2008*). In the developing epidermis, actin dynamics have been shown to directly influence active WNT signaling (*Cohen et al., 2019*) and reduced actin dynamics prevents HF placode formation (*Zhou et al., 2013*; *Ahtiainen et al., 2014*).

Hair placodes are polarized along the anterior-posterior axis by a mechanism that is known to be governed by planar cell polarity (PCP), that is the coordinated polarization of a field of cells within a tissue plane (*Aw and Devenport, 2017*; *Butler and Wallingford, 2017*). While PCP manifests in the HF, its cues are established at intercellular junctions within the E13.5 epidermal plane (*Ahtiainen et al., 2014*; *Aw et al., 2016*; *Devenport and Fuchs, 2008*; *Luxenburg et al., 2015*; *Cetera et al., 2018*). There, actin and myosin cytoskeletal dynamics orchestrate the cortical tension, cell shape remodeling and cellular rearrangements necessary to generate the epidermal PCP patterns that instruct HFs to align along the anterior-posterior axis across the tissue (*Aw et al., 2016*; *Luxenburg et al., 2015*; *Cetera et al., 2018*). While the signaling events are still unfolding, cytoskeletal networks also become asymmetrically polarized within the placode and are required for the angling of the downgrowing HF (*Cohen et al., 2019*; *Devenport and Fuchs, 2008*; *Luxenburg et al., 2015*) (*Figure 1A*).

Whether and how Rho GTPases and their effectors function in these complex actin cytoskeletal and cell junctional processes remain largely unexplored. Conventional genetic analyses have provided valuable insights into RAC1, RHOA and CDC42 Rho GTPases in skin. Postnatal roles for these Rho GTPases include stem cell maintenance, tumorigenesis, migration, wound-repair and immune-epithelial crosstalk (*Benitah et al., 2005*; *Tscharntke et al., 2007*; *Wang et al., 2010*; *Winge et al., 2016*; *García-Mariscal et al., 2018*). In addition, postnatal HF defects leading to alopecia have been reported for both RAC1 and CDC42 (*Castilho et al., 2007*; *Chrostek et al., 2006*; *Wu et al., 2006a*; *Behrendt et al., 2012*), and for CDC42, defects in basement membrane assembly, hair differentiation and WNT signaling have also been described (*Wu et al., 2006b*; *Wu et al., 2006a*). In contrast, neither CDC42 nor RHOA mutant pups show altered skin at birth, and for RHOA, mice remain aphenotypic into adulthood, suggestive of redundancy or compensation with other Rho GTPases (*Jackson et al., 2011*). The precise mechanisms by which these Rho GTPases exert the actin dynamics necessary to generate these phenotypes await further investigation. In addition, there are many other Rho GTPases and their regulators that show some expression during embryonic skin

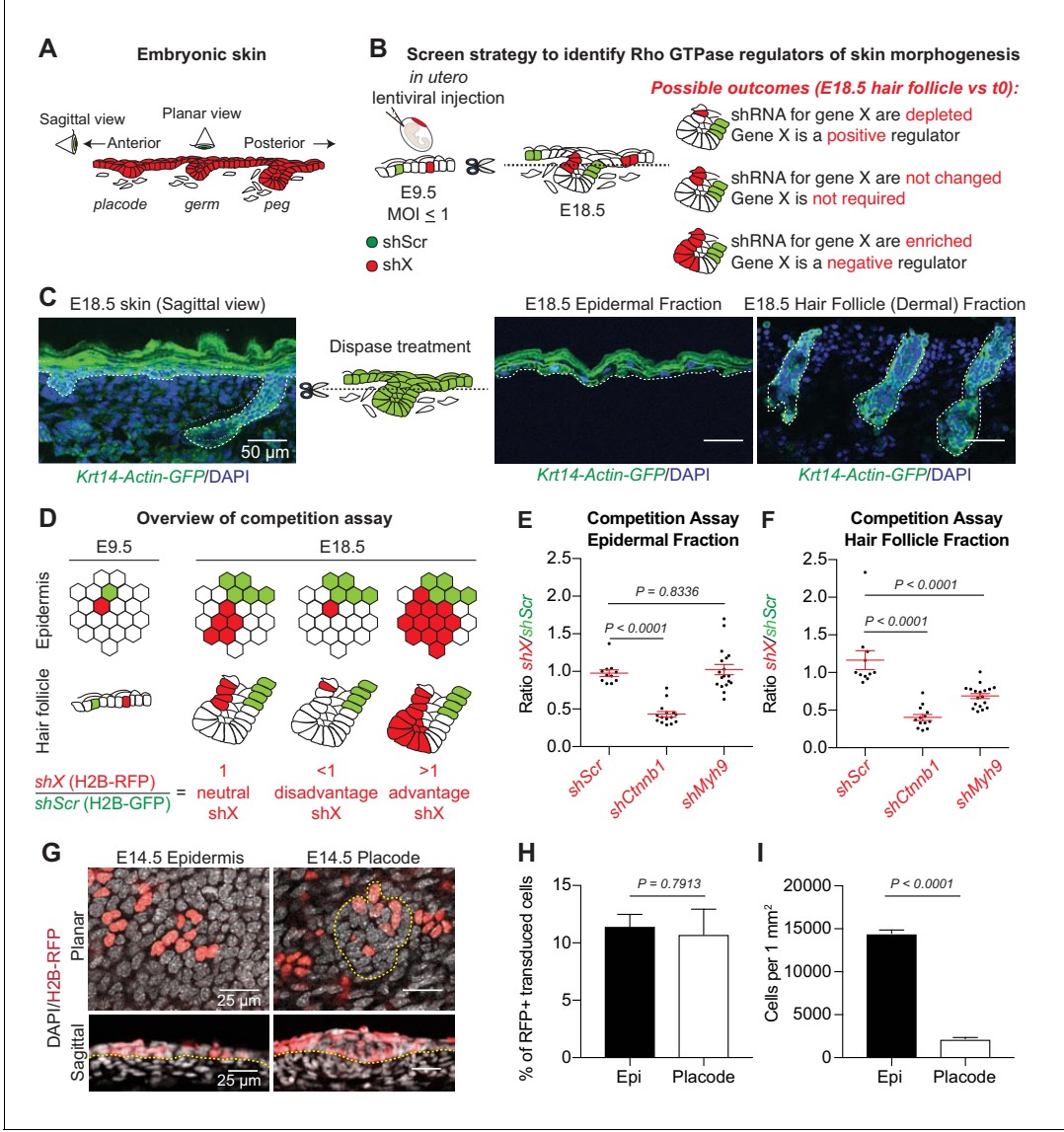

**Figure 1.** A screen strategy to identify Rho GTPase regulators of skin morphogenesis. (**A**) Schematic representation of the different staggered stages of HF development in embryonic mouse skin. (**B**) Screen strategy to identify Rho GTPase regulators of skin morphogenesis. (**C**) Dispase treatment allows for successful separation of the epidermal and HF (dermal) fractions at E18.5. Representation of n = 3 embryos. *Krt14-Actin-GFP* animals were used to visualize epidermal and HF cells. Scale bars, 50 μm. (**D**) Schematic representation of the competition assays. (**E**) Competition assay in the epidermal fraction. Error bars represent standard error of the mean (SEM) from n = 11 (*shScr/shScr*), n = 13 (*shCtnnb1/shScr*) and n = 18 embryos (*shMyh9/shScr*). Normal distribution of the data was tested using the Shapiro-Wilk test. Nonparametric unpaired two-tailed independent Mann-Whitney test was used to compare the data. (**F**) Competition assay in the HF fraction. Error bars represent SEM from n = 11 (*shScr/shScr*), n = 13 (*shCtnnb1/shScr*) and n = 18 embryos (*shMyh9/shScr*). Normal distribution of the data was tested using the Shapiro-Wilk test. Nonparametric unpaired two-tailed independent Mann-Whitney test was used to compare the data. (**G**) Top: Planar; Bottom: sagittal views of E14.5 skin showing similar level of transduction of the epidermis (left) and hair placode (right). RFP verifies transduction. Scale bars, 25 μm. (**H**) Analyses reveal that epidermal and hair placode cells are transduced in similar proportions. Error bars represent SEM from n = 15 fields from n = 3 embryos. Normal distribution of the data was tested using the Shapiro-Wilk test. Parametric unpaired two-tailed *t*-test was used to compare the data. (**I**) Placode cells represent a smaller fraction than epidermal cells in the E14.5 skin. Error bars represent SEM from n = 12 IFE fields and n = 15 placode fields from three embryos. Normal distribution of the data was tested using the Shapiro-Wilk test. Parametric unpaired two-tailed *t*-test was used to compare the data.

DOI: https://doi.org/10.7554/eLife.50226.002

The following source data is available for figure 1:

**Source data 1.** Source data related to *Figure 1E,F,H and I*.
DOI: https://doi.org/10.7554/eLife.50226.003

morphogenesis and yet whose physiological relevance and functions have been untapped (*Figure 2—figure supplement 1*) (*Rezza et al., 2016*; *Sennett et al., 2015*).

To tackle the daunting complexity of the possible Rho GTPases and their regulators that are likely to orchestrate these highly coordinated actomyosin dynamics in epidermal and HF development, we exploited the power of our ultrasound-guided microinjection system to expose E9.5 mouse embryos in utero to lentivirus and selectively transduce the single layer of skin progenitors (*Beronja et al., 2010*). Our previous screens involving pooled shRNA lentiviral libraries focused on the epidermis and scored for enrichment of library components as a consequence of altered clonal growth (*Asare et al., 2017*; *Beronja et al., 2013*; *Ge et al., 2016*; *Schramek et al., 2014*). Here, we repurposed this strategy to develop a first morphogenesis screen performed in mammals aimed at identifying Rho GTPases and their regulators that have different roles in the HF versus the epidermis. We unearthed a number of diverse functions for superfamily members in skin development. Selecting one interesting hit, the atypical Rho GTPase *Rhou*, for in depth exploration, we show how our screen can be coupled to unbiased proteomic approaches to uncover new insights into how ill-defined members of the Rho GTPases superfamily coordinate distinct types of cytoskeletal behaviors to achieve complex tissue architectures.

## Results

### A screen strategy to identify Rho GTPase regulators of skin morphogenesis

Our screen was based on the principle that cells carrying short hairpins RNA (shRNAs) that target positive regulators of HF versus epidermal development should be depleted in the growing HF, while cells carrying shRNAs against regulators whose down-regulation favors skin appendage formation should be enriched (*Figure 1B*). Our strategy further capitalized on the ease of enzymatically separating E18.5 epidermis from the dermal fraction containing HFs (*Rhee, 2006*) (*Figure 1B,C*). Since lentivirus only infects and transduces the single layer of surface epithelial progenitors, only the epithelial (i.e. HF) cells within the dermis will contain integrated lentiviral inserts, thus alleviating the need for further cell purifications before performing high throughput targeted sequencing of integrated lentiviral inserts to identify enriched and depleted shRNAs in the HFs.

The success of our previous screens focused solely on the epidermal fraction was predicated on the relatively low variation in clonal growth rates across embryonic progenitors, enabling identification of shRNAs that differentially affected growth (*Beronja et al., 2010*). To screen successfully for morphogenetic regulators of HF development, it was equally important to test and document that HF progenitors similarly undergo relatively comparable expansion during embryogenesis. We did so by performing cell competition assays followed by flow cytometry quantifications and verified that the ratio between sizes of individual cell clones expressing a shRNA against a scrambled sequence (*shScr*) and either H2B-GFP or H2B-RFP remain the same at E18.5 when compared to their starting representation (schematic in *Figure 1D*, Ratio = 1 and *Figure 1E and F*, Ratio = 1).

In addition to cellular growth, HF morphogenesis also entails cell specification, junctional remodeling and down-growth. Importantly, our competition assays were also able to identify shRNAs that perturb these features (*Figure 1E,F*). Thus in pilot tests, we observed marked depletion relative to *shScr* (Ratios <1) when we used either *shCtnnb1* targeting β-catenin, required for WNT signaling in HF specification (*Huelsken et al., 2001*), or *shMyh9* targeting Myosin IIa, which is known to be essential for HF downgrowth (*Le et al., 2016*). These shRNAs also gave the expected outcomes in the epidermal fraction: proliferation in embryonic epidermis is known to be slowed when β-catenin is defective (*Choi et al., 2013*), while Myosin IIB's redundancy with Myosin IIA as been suggested to masks in the embryo the epidermal hyperproliferation observed in its absence in adult mice (*Sumigray et al., 2012*; *Crish et al., 2013*; *Schramek et al., 2014*). These results documented the efficacy of our screen strategy to capture regulators spanning multiple aspects of skin development.

With these controls in place, we then turned to our goal of unearthing new biological functions for the understudied superfamily of Rho GTPases and their regulators. We began by building a pooled lentiviral shRNA library targeting 166 Rho GTPases and their regulators, including 20 Rho GTPases, 77 RhoGEFs, 66 RhoGAPs and 3 RhoGDIs (*Figure 2—figure supplement 1*). Our library contained ≥5 distinct shRNAs per gene, and also 20 Scr shRNAs with minimal mouse genome

homology and no effect on skin development (*Schramek et al., 2014*; *Sendoel et al., 2017*; *Yang et al., 2015*). In total, the library contained 999 independent shRNAs (*Supplementary file 1*).

For the purposes of the current study, we did not include other RAS superfamily of GTPase members to keep requisite embryo numbers for our triplicate screens manageable (<200 total). Indeed, to minimize multiplicity of infections (MOI) and ensure that epidermal progenitors receive a single shRNA, we could only infect ~15% of E9.5 surface progenitors (*Figure 2—figure supplement 2A*) (*Beronja et al., 2013*), and as our pilot testing revealed, only ~10% of transduced E9.5 epidermal progenitors contribute to HFs at E14.5 (*Figure 1G–I*). Since each E9.5 embryo harbors ~ 120,000 epidermal progenitors (*Beronja et al., 2013*), these new data implied that only ~12,000 future HF progenitors would be present at the time of injection. Thus to ensure robustness for a morphogenetic screen involving both HF and epidermal progenitors,~10X more embryos were needed over epidermal screens to have each shRNA transduced by at least 100 HF progenitors (i.e. ≥100 fold coverage) (see *Figure 2—figure supplement 2B* for further details). For our 1000 shRNA library, this meant 60 embryos per replicate.

To profile the initial composition of our lentiviral pooled library and control for variations in the representation of individual shRNAs, we infected cultured primary epidermal cells and isolated genomic DNAs 24 hr later (t0 in vitro) (*Figure 2A*). In parallel, we injected our library in triplicate in utero into the amniotic sacs of the requisite numbers of E9.5 embryos (*Figure 2—figure supplement 2B*). At E18.5, HF (tE18.5 HF) and epidermal (tE18.5 Epi) fractions were prepared from backskins, and genomic DNAs were isolated, amplified and sequenced (*Figure 2A*, *Figure 2—figure supplement 2C–E* and *Supplementary file 2*).

## RNAi screen hits reveal Rho GTPase regulators of hair follicle morphogenesis

Analysis of our sequencing data revealed a strong correlation (ranging from $R^2 = 0.9738$ to $R^2 = 0.9965$) between biological replicates (*Figure 2—figure supplement 2F–H*). Comparative analyses revealed shRNAs that were enriched or depleted in both HF and epidermal fractions relative to a) their initial representation in the library (tE18.5 HF fraction vs t0 in vitro and tE18.5 Epi fraction vs t0 in vitro) and b) the behavior of Scr controls (*Figure 2B*). Candidates were chosen for further study if ≥2 independent shRNAs each showed a ≥ 2 fold enrichment or depletion in the HF and/or epidermal fraction. Additionally, to meet our criteria, the fold change needed to be statistically significant [$P$ value < 0.05 by multiple two-tailed $t$-test and false discovery rate (FDR) method of Benjamini and Hochberch, with a FDR set at 10% between biological replicates as previously described (*Asare et al., 2017*; *Sendoel et al., 2017*)] and none of the gene's other targeting shRNAs could have the opposite effect. This optimized our chances of selecting positive hits with strong biological effects. While this also likely increased the numbers of 'false negatives', the potential for functional redundancy had already led us to focus our attention on positive hits and not negative ones. Even so, this gave us a total of 68 candidates to pursue (*Figure 2C*, *Supplementary file 3* and *Supplementary file 4*).

Seven of these genes were uniquely enriched in the epidermal fraction, making them candidates for late-stages of epidermal development (*Figure 2C*). Although some of the 35 genes enriched in both the epidermal and HF compartments were candidates for shared effects on actomyosin dynamics in these cell populations, others might affect early clonal growth in the epidermis thus impacting their future representation in the HF fraction irrespective of a function there. For the purposes of the current study, we narrowed our attention to 26 candidates whose shRNA-transduced cells were not dramatically selected for or against in the epidermal fraction but which were enriched or depleted specifically in the HF as these are strong candidates for regulating actin-mediated differences between epidermal and HF morphogenesis (*Supplementary file 5*). Four of these 26 candidates were Rho GTPases, 14 were RhoGEFs and eight were RhoGAPs (*Figure 2D*). Most were depleted in the HF fraction, suggesting a positive role in HF morphogenesis.

## Probing physiological relevance of screen hits

Of the 26 HF morphogenesis candidates, *Rac1* was in essence already validated, as prior genetic studies reported that *Rac1* was essential for HF integrity but not epidermal maintenance (*Behrendt et al., 2012*; *Chrostek et al., 2006*) (our morphogenetic screen would not have captured

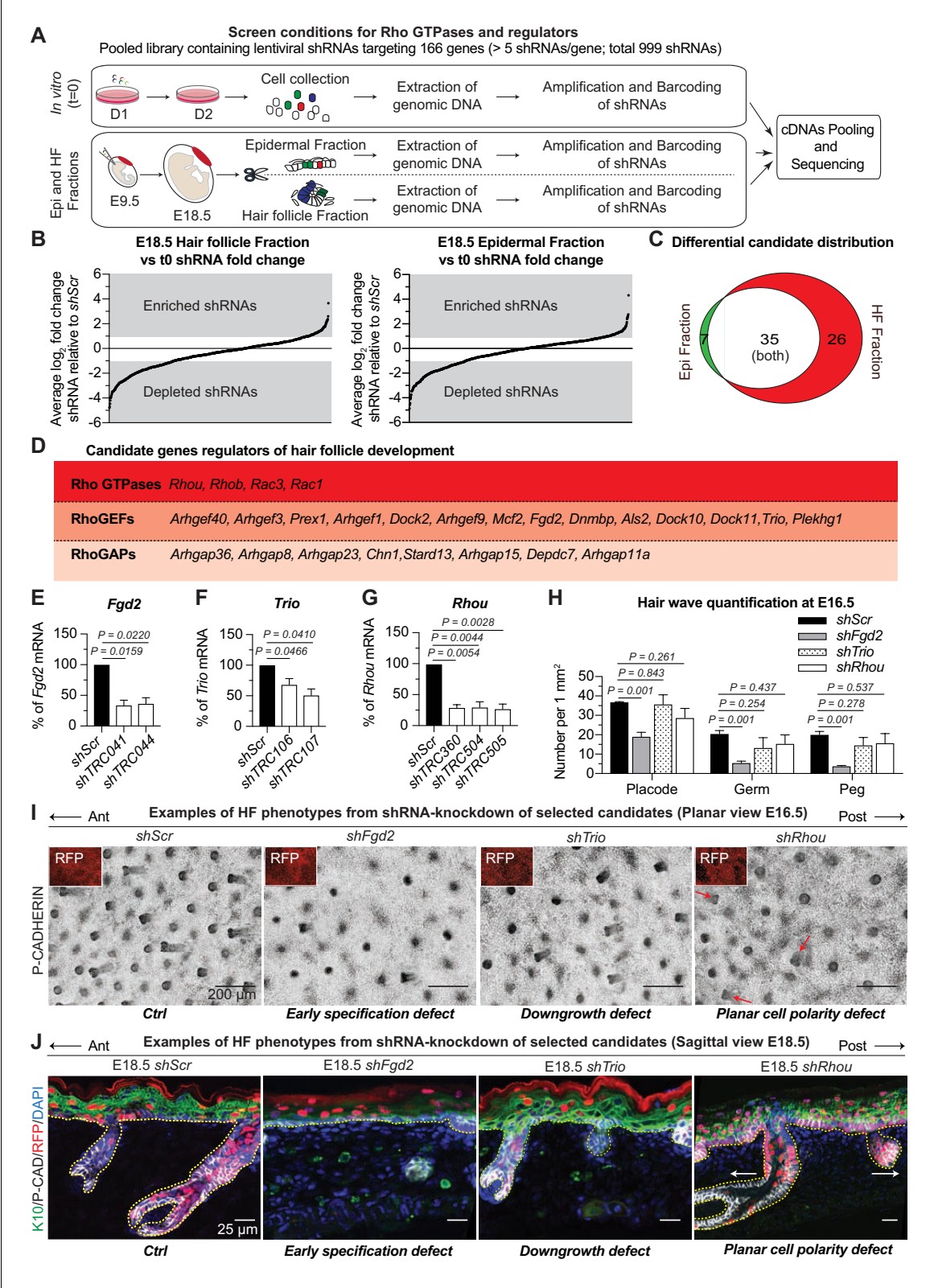

**Figure 2.** RNAi screen hits reveal Rho GTPase regulators of hair follicle morphogenesis. (**A**) Schematic representation of our in vivo screen for Rho GTPases and their regulators. (**B**) In vivo screen results showing the absolute $\log_2$ fold change for each individual shRNA present in the HF (left) and epidermal (right) fractions relative to their initial representation and normalized to 20 *Scr-shRNA* controls. (**C**) Venn diagram showing distribution of gene candidates among the epidermal and HF fractions. Candidates preferentially enriched or depleted in the epidermal fraction are highlighted in

*Figure 2 continued on next page*

Figure 2 continued

green and candidates preferentially affected in the HF fraction are highlighted in red. (D) Functional classification of the 26 candidates that were specifically either enriched or depleted in the HF fraction. (E–G) qPCR analyses in mouse keratinocytes validate shRNA-mediated knockdown of selected screen candidates. Error bars represent SEM from n = 3 experiments. Normal distribution of the data was tested using the Shapiro-Wilk test. Parametric paired two-tailed independent *t*-test was used to compare the data, (H) Quantifications of the numbers of placodes, germs and pegs from the different staggered HF waves in *shScr, shFgd2, shTrio* and *shRhou* -transduced E16.5 head skin. Error bars represent SEM from *shScr* n = 3, *shFgd2* n = 3, *shTrio* n = 3 and *shRhou* n = 5 embryos. Normal distribution of the data was tested using the Shapiro-Wilk test. Parametric independent two-tailed unpaired *t*-test was used to compare the data.(I) Examples of HF phenotypes from shRNA-knockdown of selected candidates. Z-projections of planar views from whole-mount immunofluorescence of transduced E16.5 head skins. RFP (insets) verifies transduction. Representation of n = 3 embryos for each condition. Scale bars, 200 μm. (J) Examples of HF phenotypes. Immunofluorescence of sagittal sections from transduced E18.5 skins. Representation of n = 3 embryos for each condition. Scale bars, 25 μm.

DOI: https://doi.org/10.7554/eLife.50226.004

The following source data and figure supplements are available for figure 2:

Source data 1. Source data related to *Figure 2E,F,G and H*.
DOI: https://doi.org/10.7554/eLife.50226.007
Figure supplement 1. Expression of Rho GTPases and their regulators in embryonic epidermis versus hair placodes.
DOI: https://doi.org/10.7554/eLife.50226.005
Figure supplement 2. Establishment of conditions for the screen.
DOI: https://doi.org/10.7554/eLife.50226.006

*Rac1*'s postnatal roles for example in wound-repair, stem cell function and tumorigenesis). We chose an additional three candidates for further validation: shRNAs targeting the two RhoGEF candidates, *Fgd2* and *Trio*, had been selected against in the screen; shRNAs targeting the Rho GTPase candidate, *Rhou,* were preferentially enriched in HFs. FGD2 is an understudied member of the FGD1 family of CDC42 GEFs, and mutations in this family have been associated with faciogenital dysplasia, a disease thought to originate from cellular migration defects during embryonic development (*Nakanishi and Takai, 2008*; *Pasteris and Gorski, 1999*). TRIO and its paralog KALIRIN are atypical RhoGEFs which possess two GEF domains, each with different specificity (*Schmidt and Debant, 2014*). TRIO was recently found to play a role in lens placode invagination (*Plageman et al., 2011*), a biological process that is tantalizingly similar to HF downgrowth. RHOU has a higher intrinsic nucleotide exchange rate than CDC42 in vitro and has been regarded as an atypical, constitutively active Rho GTPase (*Shutes et al., 2004*; *Hodge and Ridley, 2017*). RHOU has been implicated in development of the neural crest (*Faure and Fort, 2015*), heart (*Dickover et al., 2014*) and foregut endoderm (*Loebel et al., 2011*), all morphogenetic processes involving cell adhesion and migration.

The functions of these proteins had not been previously explored in skin, but this scant knowledge from other systems was intriguing and their appearance in our screen suggested that they had non-redundant physiological relevance. To pursue their roles in skin, we first transduced cultured epidermal cells with the individual lentiviruses that had emerged as screen hits, and employed qPCR to confirm the knockdown efficiency of their mRNAs (*Figure 2E–G*). We then infected E9.5 embryos in utero with high-titer lentiviruses harboring an *H2B-RFP* reporter gene along with a validated shRNA for each candidate. Animals with ≥80% transduction efficiency were used for phenotypic analysis of candidate-depleted skin.

The data are shown in *Figure 2H–J*. The deleterious consequences of reducing *Fgd2* expression were manifested early in HF morphogenesis, restricting the numbers of HF from all waves that are specified and that will develop. *Trio* appeared to act later in morphogenesis, as it affected HF downgrowth but not initial placode specification. Finally, *shRNA*-mediated knockdown of *Rhou* expression didn't affect the number of HF that are forming or their ability to grow in the dermis but rather revealed what appeared to be a PCP defect, as reflected by the disorganization in the anterior:posterior directionality of the HFs (*Figure 2I,J*).

Overall, the diversity of these phenotypes highlighted the power of our strategy in capturing and distinguishing important morphogenetic regulators of development, even when they are proteins within the same functional category and affecting the same tissue, in this case, the HF.

## RHOU, an unusual hit from our RNAi screen for regulators of morphogenesis

To more rigorously interrogate the power of our approach, we investigated one top screen hit in depth. We were particularly interested in RHOU, a Rho GTPase whose shRNAs were enriched on average 3x relative to the 20 scrambled controls in the HF fraction of our screen triplicates (*Figure 3A*). Moreover, in our screen, three distinct *Rhou* shRNAs exhibited this behavior, indicating a strong selective bias for downregulating this Rho during HF morphogenesis.

Labeling of living embryos with 5-ethynyl-2'-deoxyuridine (EdU) to mark S-phase cells indicated that the enrichment of *shRhou*-transduced cells was not a consequence of enhanced proliferation of cells within the HF fraction (*Figure 3B*), further distinguishing this morphogenesis screen from our prior screens aimed at growth differentials. To validate that the selective advantage of *shRhou* cells within HFs was due specifically to *Rhou* depletion, we first employed immunoblot analysis and immunofluorescence microscopy to show that RHOU was indeed targeted by the respective shRNAs (*Figure 3C,D* and *Figure 3—figure supplement 1A*). We then performed RNA sequencing to show that RHOU-deficiency did not significantly affect levels of the dozen other Rho GTPases, including closely related *Rhov*, which were expressed in skin epithelium at comparable or higher levels than *Rhou* (*Figure 3—figure supplement 1B*).

RHOU's expression pattern began to yield insights into why *shRhous* were enriched preferentially in HFs. *Rhou* mRNA expression was lower in placode than epidermal progenitors (*Figure 2—figure supplement 1A*, red box). Moreover, while RHOU protein localized cortically in embryonic basal epidermal progenitors, its expression waned during HF morphogenesis (*Figure 3E*). Pixel intensity quantification confirmed that RHOU's protein expression followed the same trend as its mRNA (*Figure 3—figure supplement 1C*). When taken together with our screen results, *Rhou's* downregulation appeared to be a necessary step in HF morphogenesis.

## RHOU: A prerequisite to establishing planar cell polarity in the skin

To follow up on our analysis of the apparent PCP phenotype observed in *shRhou505*-transduced skin (*Figure 2J*), we first verified that other *Rhou* shRNAs behaved similarly (*Figure 4A*). Quantifications revealed that by E18.5, most *Rhou* shRNA-transduced HFs from all developmental waves were misoriented and failed to align along the anterior-posterior axis of the embryo (*Figure 4A*). Most striking were some *Rhou-505* shRNA-transduced HFs that were angled $180^0$ relative to their normal orientation (*Figure 4A* images and quantifications). Other notable signs of defective PCP included loss of asymmetric polarization of P-CADHERIN, NCAM and Sonic Hedgehog (SHH) (*Figure 4—figure supplement 1A*), and an increase in follicles that grew straight downward (perpendicular to the skin surface) (*Figure 4B*).

Given RHOU's expression patterns, we next turned to tantalizing possibility that this Rho family member might be required to establish PCP cues in the basal epidermal plane. At the core of PCP is CELSR1, an atypical cadherin that localizes specifically along the anterior and posterior cell membranes of basal epidermal cells during skin development and orchestrates the distributions of other PCP components (*Devenport and Fuchs, 2008*; *Guo et al., 2004*; *Wang et al., 2006*). Perturbations in CELSR1 localization have been shown to perturb HF orientation (*Devenport and Fuchs, 2008*). In WT embryos, embryonic epidermis exhibited a temporal increase of CELSR1 planar polarization along the anterior:posterior axis (*Figure 4C,D*). In marked contrast, *shRhou* embryos showed clear perturbations in CELSR1 polarization. These defects were not attributable to differences in the expression of core PCP genes, whose levels were comparable to controls (*Figure 4E*). Rather, without RHOU, CELSR1 pattern and consequently PCP, was not properly established in the developing epidermis and this in turn prevented HFs from properly aligning.

## Beyond HF angling: Alterations in basal epidermal cell shape and actin-junctional dynamics when *Rhou* is suppressed

Defects in HF angling alone did not explain why shRNAs targeting *Rhou* surfaced in our screen. Neither our screen (*Figure 2C*) nor cell proliferation analysis (*Figure 3B*) were supportive of RHOU as a potential regulator of growth. However, since CELSR1 asymmetric localization requires actin dynamics and stabilization of adherens junctions and that PCP establishment arises concomitantly with profound cellular shapes changes in the epidermal layer (*Devenport and Fuchs, 2008*;

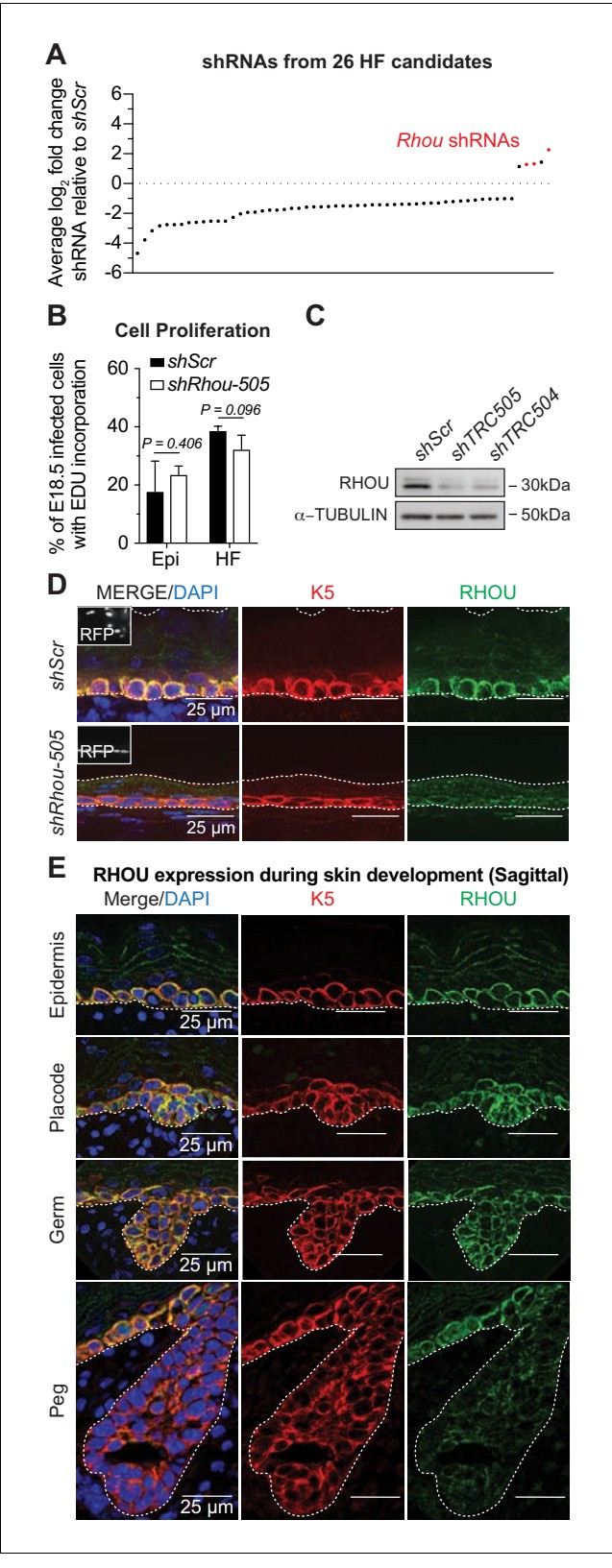

**Figure 3.** RHOU, an unusual positive hit from our RNAi screen for Rho GTPase regulators of morphogenesis. (**A**) In vivo screen results show the absolute log$_2$ fold changes of shRNAs targeting the 26 candidates specifically enriched or depleted in the HF fraction. shRNAs targeting *Rhou* are highlighted in red. (**B**) Epidermal and HF cell proliferation is not affected by RHOU loss. Quantifications of epidermal and HF EDU incorporation to quantify the

*Figure 3 continued on next page*

*Figure 3 continued*

numbers of S-phase cells. Error bars represent SEM from n = 3 *shScr* and n = 3 *shRhou-505* embryos. Normal distribution of the data was testes using the Shapiro-Wilk test. Parametric independent unpaired two-tailed *t*-test was used to compare the data. (**C–D**) RHOU immunoblot (**C**) and immunofluorescence (**D**) of E16.5 skin substantiate antibody efficacy and effective RHOU depletion by *shRNA*-mediated knockdown in vitro and in vivo. Scale bars, 25 µm. (**E**) RHOU immunofluorescence reveals expression in basal epidermal progenitors, but downregulation as HF morphogenesis proceeds. Representation of n = 3 embryos. Scale bars, 25 µm.

DOI: https://doi.org/10.7554/eLife.50226.008

The following source data and figure supplement are available for figure 3:

**Source data 1.** Source data related to *Figure 3B*.
DOI: https://doi.org/10.7554/eLife.50226.010

**Figure supplement 1.** Validation of the *shRhou* knockdown and RHOU antibody specificity.
DOI: https://doi.org/10.7554/eLife.50226.009

---

*Luxenburg et al., 2015*; *Aw et al., 2016*), we turned to the possibility that RHOU may function in this arena.

Closer inspection revealed that *shRhou* basal epidermal progenitors were flatter than normal, with elongation of the basal surface where integrin-actin junctions reside, and reduced lateral surfaces, where adherens and PCP junctions interface with the actin cytoskeleton (*Figure 3D*). Although our screen only focused on the overall growth of the epidermis and the emergence and downgrowth of the HF, and hence did not detect alterations in the stratification/terminal differentiation program of the epidermis, these alterations modestly affected the overlying layers. This resulted in a thinner epidermis with reduced suprabasal keratin 10 (K10) and LORICRIN, and induction of keratin 6 (K6), a marker of aberrant epidermal behavior (*Figure 4—figure supplement 1B,C*). By E18.5, however, the barrier was largely intact (*Figure 4—figure supplement 1D*).

To further explore the hypothesis that RHOU might be acting by regulating actin-associated cell-cell junction and cell shape dynamics, we turned to whole mount immunofluorescence (*Figure 5A* and *Figure 5—figure supplement 1A*). Imaging of the basal plane of normal E14.5 epidermis showed a honeycombed pattern of cells displaying smooth cell-cell borders of P-CADHERIN immunolabeling (left panels). Quantification of pixel intensity across individual cells revealed that P-CADHERIN intensity became increasingly stronger at cell borders between E14.5 and E15.5 (right panels). This correlated with a reorganization of the filamentous actin into a cortical network as observed by the increased in pixel intensity of F-ACTIN at cell borders over time. By contrast, the epidermal sheet of E14.5 *shRhou*-transduced embryos showed considerably greater disorganization, with immature intercellular borders, as reflected by a broader distribution of P-CADHERIN and a failure to properly reorganize filamentous actin into a cortical network reflected by the increased in pixel intensity of F-ACTIN in the cytosol of *shRhou*-transduced cells.

While control basal epidermal cells were more cuboidal at E15.5 than earlier in development (Z panels), basal cells from either *shRhou-505* or *shRhou-504* were flatter, a feature that accounted for their larger appearance in planar views, and substantiated our data from sagittal sections (*Figure 5A Figure 5A*; *Figure 5—figure supplement 1A* and *Figure 3D*). Shape analysis of these cells within the E14.5 and E15.5 basal epidermal sheet revealed that indeed, the average area along the basement membrane of *shRhou* cells was larger than *shScr* counterparts, and this was accompanied by a marked decrease in their basal cell height, a feature still present at E16.5 (*Figure 5B* and *Figure 5—figure supplement 1B*). Finally, these cell shape differences resulted in a decrease in basal cell density, which lessened concomitant with the natural increase in density that occurs during epidermal maturation (*Figure 5B* and *Figure 5—figure supplement 1B*). Overall, our findings pointed to a functional requirement for RHOU in promoting (actin-dependent) adherens junction formation, diminishing the integrin-rich basal surface and generating the cortical actin network that is necessary for columnar cell shape, CELSR1 orientation and PCP establishment in the epidermal plane (*Luxenburg et al., 2015*). Consistent with this notion, the loss of RHOU was accompanied by an increase in the phosphorylation of myosin II light chain, expected from a more robust actomyosin network (*Figure 5C* and *Figure 5—figure supplement 1C*).

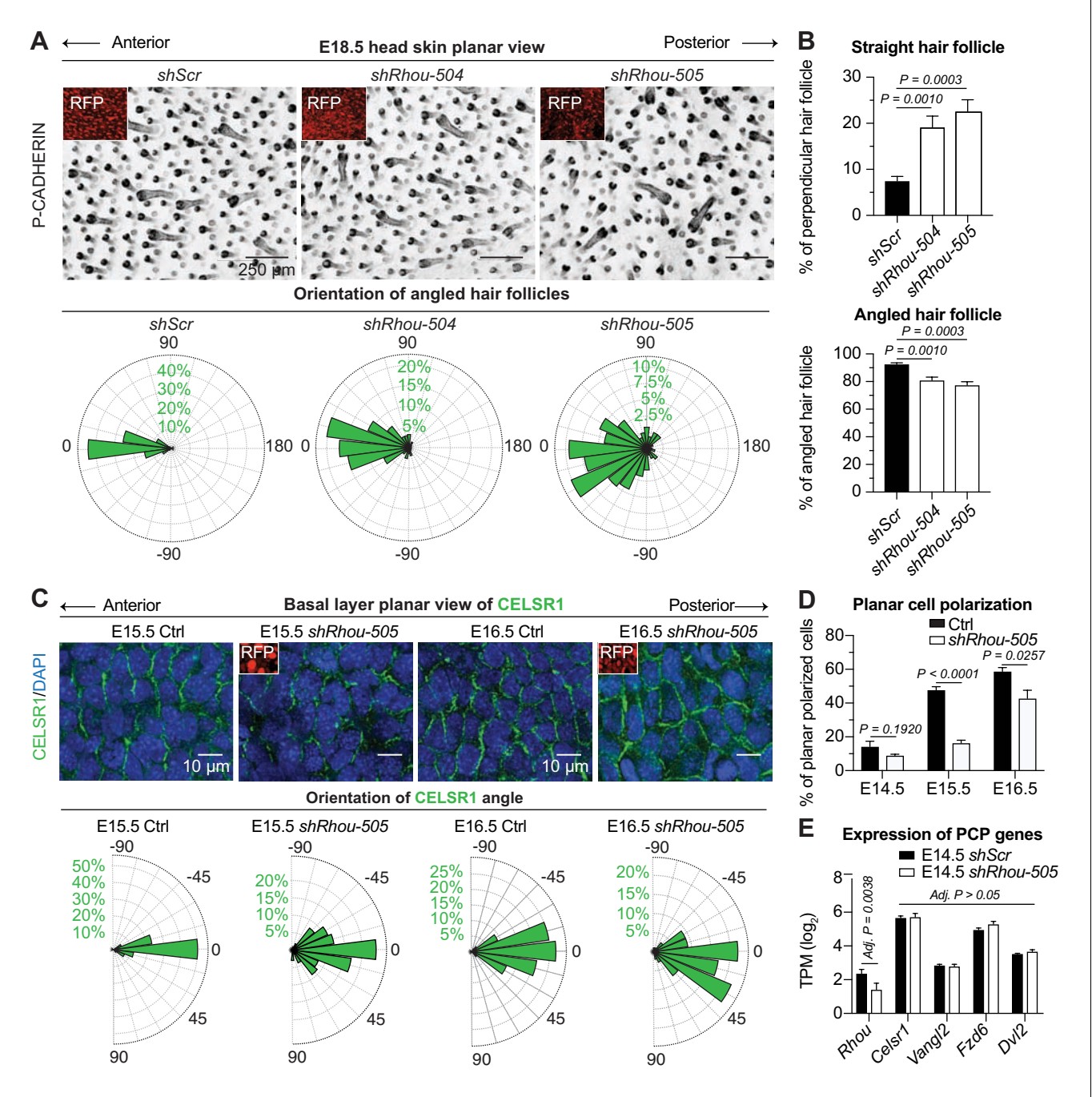

**Figure 4.** RHOU: a prerequisite to establishing planar cell polarity in the skin. (A) RHOU-depleted skins have misoriented HFs. (Top): Z-projections of planar views from whole-mount P-CADHERIN immunofluorescence of *shScr* and *shRhou*-transduced E18.5 head skins. RFP (insets) verifies transduction. Representation of n = 3 embryos per shRNA. Scale bars, 250 μm. (Bottom): Frequency distribution of HF orientations. *shScr* n = 471 HFs; *shRhou-504* n = 470 HFs; *shRhou-505* n = 484 HFs from n = 3 embryos. *P* =< 0.0001 (*shScr* vs *shRhou-504*), p<0.0001 (*shScr* vs *shRhou-505*) (Kolmogorov-Smirnov test). (B) Quantifications of the average percentage of perpendicular (top panel) and angled (bottom panel) HFs per mouse. *shScr* n = 527, *shRhou-504* n = 407, *sRhou-505* n = 552 HFs from n ≥ 3 embryos. Normal distribution of the data was tested using the Shapiro-Wilk test. Parametric independent two-tailed unpaired *t*-test was used to compare the data. (C,D) Planar cell polarization during skin development. CELSR1 immunofluorescence was used to measure the anterior-posterior polarization of each cell. (C): (Top): Planar views from CELSR1 whole-mount immunofluorescence of transduced E15.5 and E16.5 head skins. Shown are the midplanes of the basal cell layer. Representations from n = 3 embryos per condition. Scale bars, 10 μm. (Bottom): Frequency distribution of the orientation of CELSR1 polarity along the anterior:posterior axis. E15.5 Ctrl = 184, E15.5 *shRhou-505* = 81, E16.5 Ctrl = 319 and E16.5 *shRhou-505* = 158 cells from ≥3 embryos. p=0.0033 (E15.5 Ctrl vs *shRhou-505*), *P* =<0.0001 (E16.5 Ctrl vs *shRhou-505*). Kolmogorov-Smirnov test was used to compare the distribution. (D) Temporal changes in the percentages of planar polarized cells. Error bars represent SEM from E14.5

*Figure 4 continued on next page*

*Figure 4 continued*

n = 3, E15.5 n = 4 and E16.5 n = 4 embryos for each time point;>100 cells per embryo per conditions where analyzed. Normal distribution of the data was tested using the Shapiro-Wilk test. Parametric unpaired two-tailed independent *t*-test was used to compare the data. (**E**) Differential gene expression analysis (DESeq2) of transcriptome profiling from E14.5 transduced epidermal cells. Note that *shRhou* did not affect the levels of core PCP transcripts.

DOI: https://doi.org/10.7554/eLife.50226.011

The following source data and figure supplement are available for figure 4:

**Source data 1.** Source data related to *Figure 4A,B,C and D*.

DOI: https://doi.org/10.7554/eLife.50226.013

**Figure supplement 1.** RHOU controls the proper shape of early basal epidermal progenitors which in turn is needed for tissue development.

DOI: https://doi.org/10.7554/eLife.50226.012

## Proximity-dependent RHOU interactome unveils a bias for proteins known to shuttle between adherens junctions and focal adhesions

To probe more deeply into the molecular basis of RHOU's effects on cell-cell junction and shape dynamics, we turned to an in vitro system of cultured primary neonatal skin epidermal keratinocytes. We first tested whether in these cells, RHOU regulates adherens junction formation and actin dynamics as it did in vivo. Indeed, upon shifting from low to high calcium medium to activate cell-cell junction assembly, RHOU-deficient epidermal cells exhibited clear delays in adherens junction formation (*Figure 5—figure supplement 2A*). After 24 hr, however, intercellular junctions were indistinguishable from WT (not shown), indicating that RHOU-deficiency did not block intercellular junction formation per se. Rather, RHOU appeared to function in supporting nascent adherens junctions at early stages of epithelial sheet formation, where adherens junction-linked actin dynamics are required (*Vasioukhin et al., 2000*).

Upon examining individual cells, it became clear that loss of RHOU resulted in an increase in F-ACTIN stress fibers and integrin-mediated focal adhesion size and length, features of reduced focal adhesion turnover (*Figure 5—figure supplement 2B–D*). This was intriguing, as it began to illuminate further why basal epidermal progenitors lacking RHOU in vivo were on the one hand, flatter with enhanced basal surface along the ECM-rich basement membrane and on the other, less columnar with reduced apico-lateral surfaces along sites of intercellular contact.

Given the similarities in RHOU's in vitro and in vivo effects, we turned to proximity-dependent biotin identification ('BioID') (*Roux et al., 2012*) to position RHOU at the core of signaling pathways and gain further molecular insights into its function. The BioID method relies on fusion of a promiscuous biotin ligase (BirA*) to a protein of interest, such that upon expression in keratinocytes, endogenous proteins in proximity will be biotinylated on available lysine residues and recovered through streptavidin pull-down (*Figure 6—figure supplement 1A*). This method detects transient protein interactions and can accommodate harsh lysis conditions that often prevent protein partner identification, but which are necessary for epidermal keratinocytes. For our purposes here, we generated a doxycycline-inducible MYC-tagged biotin ligase (MYC-BioID2) (*Kim et al., 2016*) fused with RHOU or a similarly sized GFP control. Each 'bait' was packaged into lentiviral constructs harboring an H2B-RFP reporter for identifying transduced cells (*Figure 6A*).

Primary epidermal progenitors cultured from P0 *K14rtTA* transgenic mouse skin were then transduced with either *Myc-BioID2-Rhou* or *Myc-BioID2-GFP* lentiviruses and stable transduced lines were generated. By anti-MYC immunoblot and immunofluorescence analyses, the two doxycycline-induced baits were expressed at similar levels, and appropriate sizes (*Figure 6B* and *Figure 6—figure supplement 1B*). By immunofluorescence, MYC-BioID2-RHOU, but not MYC-BioID2-GFP, localized to focal adhesions, marked by VINCULIN, in normal growth conditions and redistributed to intercellular borders, marked by E-CADHERIN, upon calcium-induced activation of adherens junction formation and terminal differentiation (*Figure 6—figure supplement 1C*). As expected, both fusion proteins activated biotinylation of endogenous proteins in the presence of biotin (*Figure 6—figure supplement 1D*). With these controls in hand, we administered biotin 5d after bait induction, and following streptavidin affinity purification, biotinylated proteins were identified by mass spectrometry.

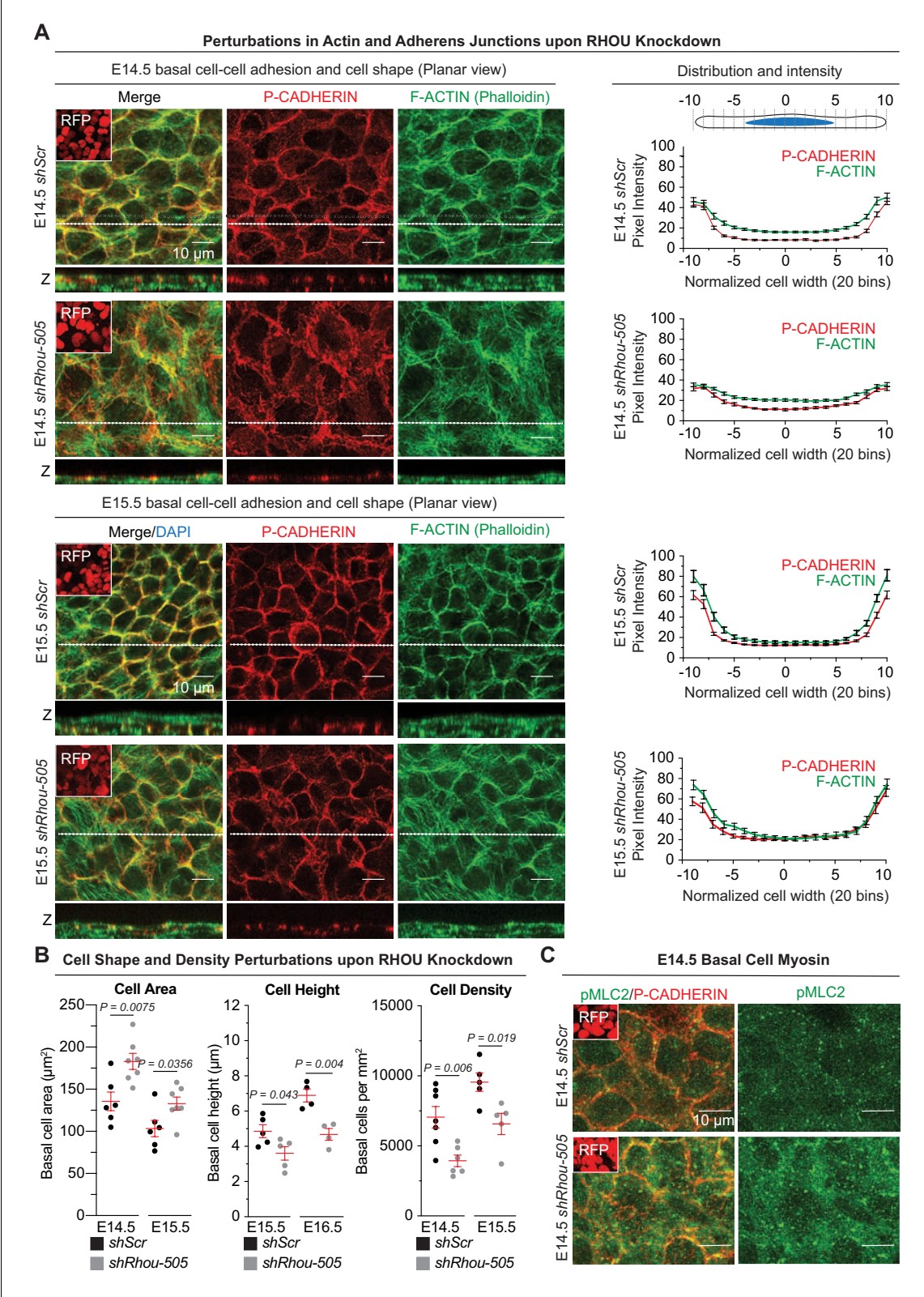

**Figure 5.** Functions of RHOU in orchestrating cellular junction dynamics within the developing epidermis. (A) Perturbations in cortical distribution of F-ACTIN and P-CADHERIN upon *shRhou* knockdown. Planar views from whole-mount immunofluorescence of transduced E14.5 (Top) and E15.5 (Bottom) head skins. P-CADHERIN marks adherens junctions, Phalloidin marks F-ACTIN, RFP verifies transduction. Shown are representative images from the midplanes of the basal cell layers. Representation of n = 3 embryos. Scale bars, 10 µm. (Right) Pixel intensity measurements across the

*Figure 5 continued on next page*

Figure 5 continued

diameter of basal progenitors. Note the broader distribution of P-CADHERIN and the increase intensity of F-ACTIN in the cytoplasm of *shRhou*-transduced cells. (B) Cell measurements reveal perturbations upon *Rhou* knockdown in the area, height and density of basal progenitors within the epidermal plane. For cell area, error bars represent SEM from E14.5 *shScr* n = 6, E14.5 *shRhou-505* n = 7, E15.5 *shScr* n = 6 and E15.5 *shRhou-505* n = 7 embryos. Normal distribution of the data was determined using the Shapiro-Wilk test. Parametric unpaired independent two-tailed *t*-test was used to compare the data. For quantifications of basal cell height, error bars represent SEM from E15.5 *shScr* n = 5, E15.5 *shRhou-505* n = 5, E16.5 *shScr* n = 4 and E16.5 *shRhou-505* n = 4 embryos. Normal distribution of the data was determined using the Shapiro-Wilk test. Parametric unpaired independent two-tailed *t*-test was used to compare the data. For quantifications of basal cell densities, error bars represent SEM from E14.5 *shScr* n = 7, E14.5 *shRhou-505* n = 6, E15.5 *shScr* n = 5 and E15.5 *shRhou-505* n = 5 embryos. Normal distribution of the data was determined using the Shapiro-Wilk test. Parametric unpaired independent two-tailed *t*-test was used to compare the data. (C) RHOU depletion increases the level of Myosin II light chain phosphorylation. Planar views from whole-mount immunofluorescence of transduced E14.5 skins. P-CADHERIN and pMLC2 immunofluorescence. RFP verifies transduction. Shown are representative images from the midplanes of the basal cell layers from n = 3 embryos. Scale bars, 10 μm.

DOI: https://doi.org/10.7554/eLife.50226.014

The following source data and figure supplements are available for figure 5:

**Source data 1.** Source data related to *Figure 5A and B*.

DOI: https://doi.org/10.7554/eLife.50226.017

**Figure supplement 1.** RHOU depletion via *shRhou-504* results in analogous defect in orchestrating cellular junction dynamics within the developing epidermis.

DOI: https://doi.org/10.7554/eLife.50226.015

**Figure supplement 2.** RHOU regulates epidermal cell adhesion and junction dynamics.

DOI: https://doi.org/10.7554/eLife.50226.016

Proteins were considered part of RHOU's proximity interactome if they were identified only in biological replicates of MYC-BioID2-RHOU expressing progenitors and not their MYC-BioID2-GFP expressing counterparts. Ninety-five proteins met these criteria (*Supplementary file 6*). The frequency of each protein's detection in the interactome was illustrated by the size of each word in *Figure 6C*. This RHOU interactome highlighted not only a handful of proteins that were previously found to be associated with this atypical Rho GTPase (*Saras et al., 2004*; *Shutes et al., 2004*) but also a considerable cohort of hitherto unreported putative partners. Gene ontology analysis (*Figure 6D*) revealed a RHOU interactome enriched for proteins involved in 'adherens junctions', including p120-catenin (Ctnnd1), a known RHOA-interacting protein (*Perez-Moreno et al., 2006*) and 'focal adhesions', including paxillin, integrin β1 and tensin-4, as well as NCK1, an adaptor protein required for Rho GTPase activation and cell migration (*Saras et al., 2004*), which was previously reported as an interactor of RHOU (*Risse et al., 2013*). An outlier was PAR6, which did not surface in our interactome, despite its expression in primary keratinocytes (*Figure 6—figure supplement 1E*) and previous reports of its binding to RHOU (*Brady et al., 2009*). Its absence is likely rooted in our conditions used for analysis, where keratinocytes did not display tight junctions, the site of this described interaction. Overall, the RHOU interacting proteins we unearthed were in good agreement with prior in vitro studies that have implicated RHOU in focal adhesion turnover and cell migration (*Chuang et al., 2007*; *Dart et al., 2015*). These findings were also consistent with our in vitro and in vivo data on epidermal progenitors.

Given RHOU's dual and opposing effects on the epidermis' intercellular and cell-substratum borders, we were especially intrigued by the prominence in RHOU's interactome of the p21-activated serine-threonine kinases, PAK1 and PAK2, and the guanine nucleotide exchange protein ARHGEF7 (β-PIX). These proteins are known to shuttle as a complex between adherens junctions, where tyrosine kinase receptor activity is silent and focal adhesions, where it is active (*Livshits et al., 2012*; *Plutoni et al., 2016*). Interestingly, EGFR was also present in our interactome (*Supplementary file 6*). Overall, these associations began to shed light on RHOU as a GTPase functioning in this pathway, and prompted us to probe further into their relation to RHOU.

PAK proteins are known to bind RHOU (*Shutes et al., 2004*), and RHOU can activate PAK1 (*Tao et al., 2001*). In primary keratinocytes and during epidermal development, PAK2 is the most highly expressed PAK1/2/3 protein and gene (*Figure 6E*, left and right respectively) that likely accounts for why PAK2 was more abundant than PAK1 in our proteomic analysis (*Figure 6C*). Notably, overexpression of RHOU resulted in PAK1/2 activation in keratinocytes (*Figure 6F*) while depletion of RHOU resulted in its reduction (*Figure 6G*). Moreover, we found that activation of PAK1/2

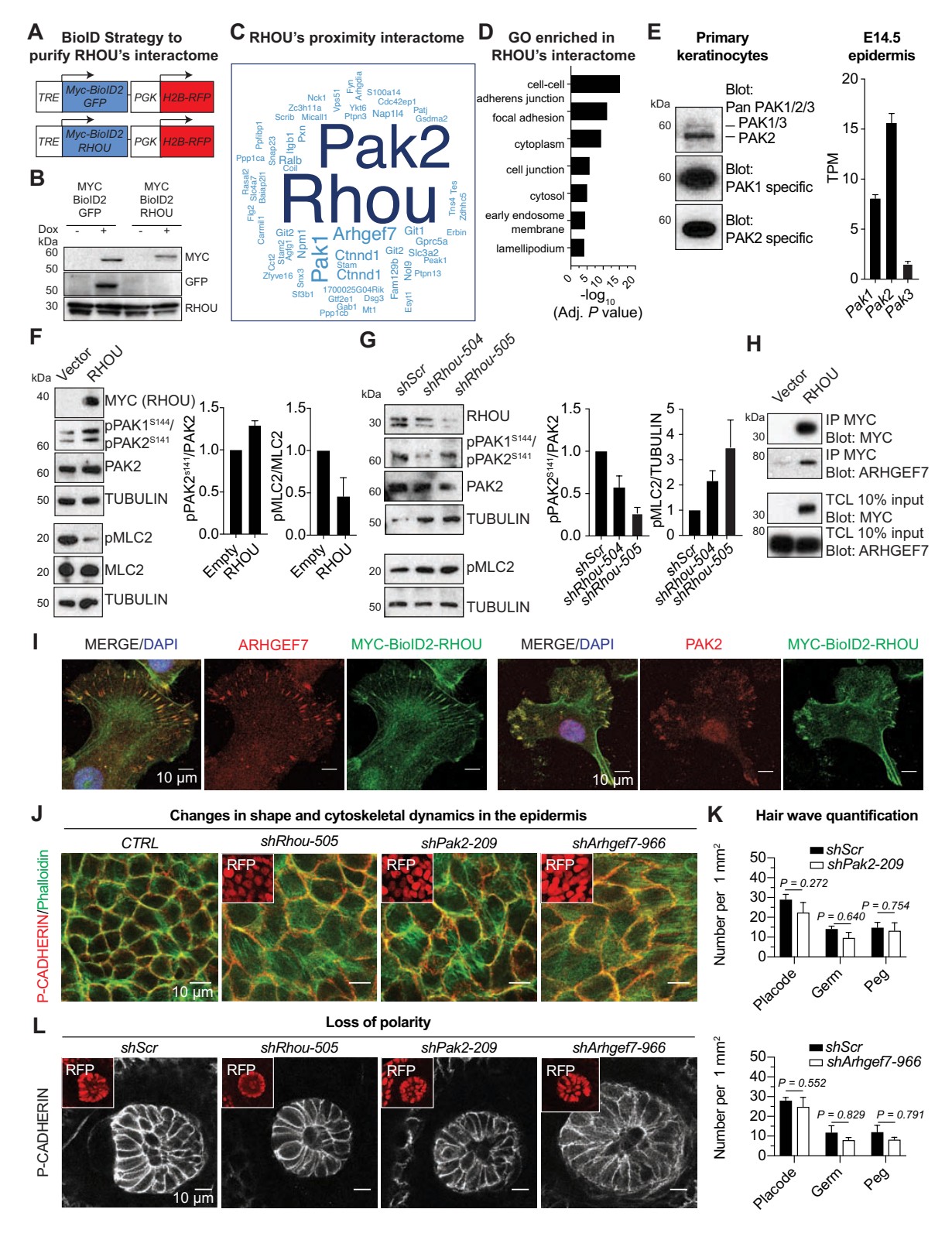

**Figure 6.** RHOU as a coordinator of cell-cell and cell-substratum junctions. (**A**) BioID2 constructs used to generate stably transduced lines from primary *Krt14-rtTA* mouse keratinocytes. TRE, tetracycline regulatory element used with a minimal promoter to drive expression of MYC-BioID2-GFP or MYC-BioID2-RHOU proteins. PGK, constitutively active promoter, used to drive expression of H2B-RFP to mark transduced cells. (**B**) Immunoblot of lysates from *Krt14rtTA*[+] keratinocytes transduced with *Tre-Myc-BioID2-GFP-pgk-H2B-RFP* or *Tre-Myc-BioID2-Rhou-pgk-H2B-RFP* lentiviruses and then treated

*Figure 6 continued on next page*

*Figure 6 continued*

for 2 days with doxycycline. Immunoblots were probed with MYC, GFP and RHOU antibodies. (C) Cloud analysis representing RHOU's proximity interactome. Font size represents frequency of most abundant proteins in the interactome. (D) Gene ontology (GO) analysis reveals a significant enrichment for 'cell-cell adherens junction' and 'focal adhesion' proteins in RHOU's proximity interactome. (E) Of the three PAK family members, PAK2 is the most highly expressed at both protein (left) and transcript (right) levels in primary keratinocytes and epidermis. (Left): Immunoblot of lysates from *Krt14rtTA*[+] mouse keratinocytes showing the higher expression of PAK2. Immunoblots were probed with a PAN PAK1/2/3, a specific PAK1 and a specific PAK2 antibody. (Right): Transcriptome profiling from E14.5 transduced epidermal cells revealed *Pak2* as the most highly expressed *Pak1/2/3* gene. (F) Overexpression of RHOU promotes PAK2 activation and decreases phosphorylation of MLC2. (Left): Primary keratinocytes were transfected with either empty vector or *Myc-Rhou*. Immunoblot of lysates were probe with MYC, pPAK1$^{Ser144}$/pPAK2$^{Ser141}$, PAK2, pMLC2, MLC2 and TUBULIN antibodies. (Right): Quantification of protein lysates. Data are represented as SEM from n = 3 experiments. (G) Depletion of RHOU reduces the activation of PAK2 and promotes the phosphorylation of MLC2. Primary keratinocytes were transfected with *shScr*, *shRhou-504* or *shRhou-505*. Immunoblots of lysates were probed with RHOU, pPAK1$^{Ser144}$/pPAK2$^{Ser141}$, PAK2, pMLC2 and TUBULIN antibodies. (Right): Quantification of protein lysates. Data are represented as SEM from n = 3 experiments for PAK2 level of phosphorylation and n = 4 for pMLC2 level of phosphorylation. (H) ARHGEF7 co-immunoprecipitates (co-IP) with MYC-RHOU in primary keratinocytes. Immunoblot of lysates and co-IP transfected with either an empty vector or *Myc-Rhou*. Immunoblots were probed with MYC and ARHGEF7 antibodies. (I) Immunofluorescence showing the co-localization of MYC-BioID2-RHOU with ARHGEF7 and PAK2 at focal adhesions in primary keratinocytes. Scale bars, 10 μm (J–L) Similarities between RHOU, PAK2 and ARHGEF7 deficiency phenotypes in skin. (J) Planar views from whole-mount immunofluorescence of transduced E15.5 headskin epidermis. P-CADHERIN marks adherens junctions; Phalloidin marks F-ACTIN; RFP verifies transduction. Shown are representative images from the midplanes of the basal cell layers. Representation from n = 3 embryos. Scale bars, 10 μm. (K) Quantifications of the numbers of placodes, germs and pegs from the different staggered HF waves in *shScr*, *shPak2-209* and *shArhgef7-966* transduced E16.5 head skin. Error bars represent SEM from *shScr* for *Pak2* n = 5, *shPak2-209* n = 5, *shScr* for *Arhgef7* n = 4 and *shArhgef7* n = 4 embryos. Normal distribution of the data was tested using the Shapiro-Wilk test. Parametric independent unpaired two-tailed *t*-test was used except for the comparison of *shScr* germ vs *shArhgef7* germ and *shScr* peg vs *shArgef7* peg for which a Mann-Whitney test was used. (L) Planar view of P-CADHERIN immunofluoresence of transduced hair peg imaged at the midplane and showing the loss of planar polarized distribution in the absence of RHOU, PAK2 and ARHGEF7. Representation from n = 3 embryos. Scale bars, 10 μm.

DOI: https://doi.org/10.7554/eLife.50226.018

The following source data and figure supplements are available for figure 6:

**Source data 1.** Source data related to *Figure 6F,G and K*.
DOI: https://doi.org/10.7554/eLife.50226.021
**Figure supplement 1.** Overview of BioID strategy and validation of fusion constructs.
DOI: https://doi.org/10.7554/eLife.50226.019
**Figure supplement 2.** Similarities between RHOU, PAK2 and ARHGEF7 deficiency phenotypes in skin.
DOI: https://doi.org/10.7554/eLife.50226.020

inversely correlated with phospho-myosin light chain 2, consistent with the view that RHOU functions by enhancing actin dynamics and focal adhesion turnover.

Among the uncharacterized proximity partners, we used immunoprecipitation and confirmed ARHGEF7's interaction with RHOU as well (*Figure 6H*). We further demonstrated the in vitro co-localization of these three proteins in primary epidermal keratinocytes (*Figure 6I*). Turning to physiological relevance, we knocked down both *Arhgef7* and *Pak2* in skin epithelium in utero. Using a newly confirmed *shArhgef7* (*Figure 6—figure supplement 2A*) and a previously established *Pak2 shRNA* (*Livshits et al., 2012*), we found that the loss of these proteins yielded delays in adherens junction formation and cortical actin reorganization in the growing epidermal plane (*Figure 6J and Figure 6—figure supplement 2B*), without appreciably affecting the numbers of HFs that formed (*Figure 6K*). Moreover, as we observed with RHOU loss, PCP defects arose in the absence of PAK2 and ARHGEF7 and were manifested by the increase of HFs that grew straight downward (*Figure 6L*). Altogether, our studies suggested a model whereby RHOU functions to promote cytoskeletal dynamics that favor adherens junction formation and, RHOU minimizes the integrin-rich basal surface.

## Functional implications of RHOU on cell shape dynamics within the developing hair follicle

Equipped with these new insights into RHOU's function in the epidermis, we focused on ferreting out why loss of RHOU is favored in HF morphogenesis. Based upon our findings in the epidermis, we posited that the process must benefit from reducing intercellular contacts and enhancing cell-substratum adhesion. To begin to test this possibility, we engineered doxycycline-inducible MYC-RHOU embryos and overrode RHOU's normal downregulation during HF morphogenesis (*Figure 7A*

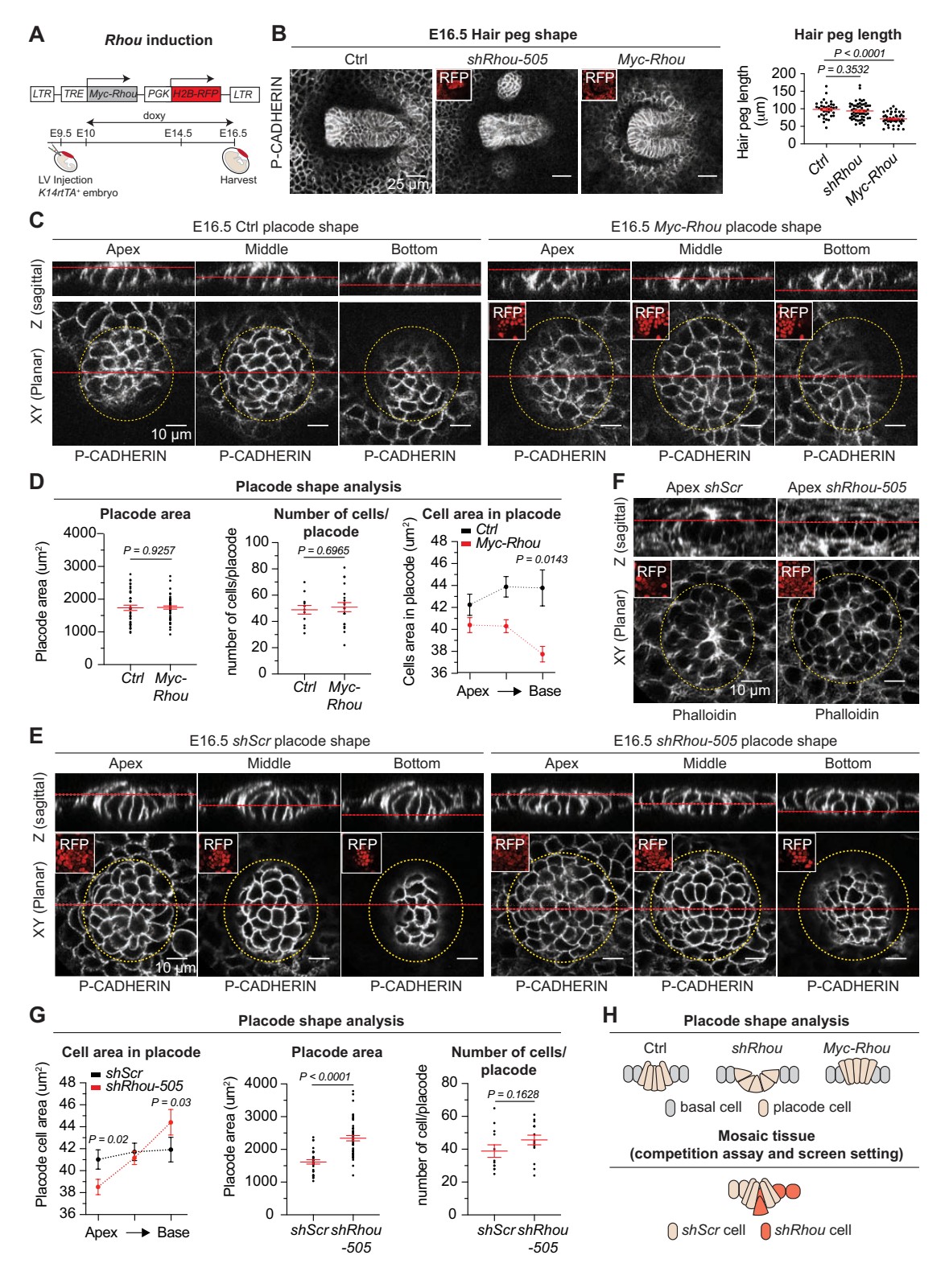

**Figure 7.** Functional implications of RHOU in cellular shape dynamics involved in hair follicle morphogenesis. (**A**) Strategy to induce *Rhou* expression in embryonic skin. (**B**) Sustained RHOU in developing HFs perturbs their downgrowth. (Left panel): Planar views from whole-mounts of E16.5 Ctrl, *shRhou* and *Myc-Rhou* transduced hair pegs. Representation of n > 3 embryos. Scale bars, 25 µm. (Right panel): Quantification of hair peg length. Error bars represent SEM from E16.5 Ctrl n = 36, *shRhou-505* n = 58 and *Myc-Rhou* n = 50 hair pegs from ≥3 embryos. Normal distribution of the data was tested

*Figure 7 continued*

using the Shapiro-Wilk test. Parametric independent two-tailed unpaired *t*-test was used to compare the data. (**C**) Perturbations in placode shape upon RHOU overexpression. Views from whole-mount of E16.5 *Ctrl* and *Myc-Rhou* transduced placodes at the indicated representative plane. Representation of n = 3 embryos. In all images, RFP insets verify transduction of the tissue. Scale bars for all images, 10 μm. (**D**) Placode shape analysis. (Left panel): Quantification of placode areas. Error bars represent SEM from E16.5 Ctrl n = 39 and *Myc-Rhou* n = 49 placodes from >3 embryos. Normal distribution of the data was tested using the Shapiro-Wilk test. Parametric unpaired two-tailed *t*-test was used to compare the data. (Middle panel): Quantifications of the number of cells per placode. Error bars represent SEM from E16.5 Ctrl n = 12 and *Myc-Rhou* n = 20 placodes from >3 embryos Normal distribution of the data was determined using the Shapiro-Wilk test. Parametric unpaired two-tailed *t*-test was used to compare the data. (Right panel): Quantifications of the cell area in placodes at the indicated plane. Error bars represent SEM from E16.5 Ctrl Top n = 453, Ctrl Middle n = 443, Ctrl Bottom n = 396, *Myc-Rhou* Top n = 1053, *Myc-Rhou* Middle n = 1111 and *Myc-Rhou* Bottom n = 1026. Normal distribution of the data was determined using the Shapiro-Wilk test. Nonparametric unpaired two-tailed Mann Whitney test was used to compare the data. Note the marked reduction in the basal surface of RHOU-sustained placode cells. (**E**) Cellular remodeling in the early steps of HF morphogenesis is perturbed in the absence of RHOU. Views from whole-mount immunofluorescence of transduced placodes at the indicated representative plane. Representation of n > 3 embryos. (**F**) F-ACTIN (as detected by Phalloidin) is normally enriched at the placode apex, but is perturbed in the absence of RHOU. Views from whole-mount of transduced placodes. Representation of >3 embryos. (**G**) Placode shape analysis. (Left panel): Quantifications of the cell areas within the placodes at the indicated planes. Error bars represent SEM from E16.5 *shScr* Top n = 491, *shScr* Middle n = 467, *shScr* Bottom n = 449, *shRhou-505* Top n = 627, *shRhou-505* Middle n = 685 and *shRhou-505* Bottom n = 506 of the cells from >3 embryos. Normal distribution of the data was determined using the Shapiro-Wilk test. Nonparametric unpaired two-tailed Mann-Whitney test was used to compare the data. (Middle panel): Quantifications of the placode area reveals a broadening of *shRhou*-transduced placodes. Error bars represent SEM from E16.5 *shScr* n = 31 and *shRhou-505* n = 48 placodes from n = 4 embryos. Normal distribution of the data was determined using the Shapiro-Wilk test. Parametric unpaired two-tailed *t*-test was used to compare the data. (Right panel): Quantifications of the numbers of cells per placode. Error bars represent SEM from E16.5 *shScr* n = 12 and *shRhou-505* n = 15 placodes from n > 3 embryos. Normal distribution of the data was tested using the Shapiro-Wilk test. Parametric unpaired two-tailed *t*-test was used to compare the data. (**H**) Model for how the natural changes in RHOU expression during HF morphogenesis drive the cell shape dynamics needed for invagination and downgrowth (top) and how in a mosaic setting as it is the case in our screen RHOU-depleted cells might have an advantage (bottom).
DOI: https://doi.org/10.7554/eLife.50226.022

The following source data and figure supplement are available for figure 7:

**Source data 1.** Source data related to *Figure 7B,D and G*.
DOI: https://doi.org/10.7554/eLife.50226.024
**Figure supplement 1.** Schematic representation of RHOU's role during skin morphogenesis.
DOI: https://doi.org/10.7554/eLife.50226.023

and *Figure 7—figure supplement 1A*). While HFs were still specified in similar number upon sustained RHOU expression (*Figure 7—figure supplement 1B*), their downward growth was markedly impaired in comparison to control or *Rhou*-depleted HFs (*Figure 7B* and *Figure 7—figure supplement 1C*).

Probing deeper, the area occupied by individual placodes and total numbers of cells per placode remained largely unaffected, but cells exhibited a marked narrowing of their basal domain when RHOU was sustained (*Figure 7C* Z-plane images and *Figure 7D*). Thus, while control columnar hair bud cells converted to a conical shape to achieve invagination, RHOU-sustained buds failed to reduce their apical cell surface and undergo this transition.

Conversely, upon loss of RHOU, placode cells shifted to an exacerbated conical shape in comparison to wild-type placode cells, typified by a strikingly expanded basal cell domain, a reduction in apical F-actin and a corresponding shrinking of the apical area compared to control cells (*Figure 7E–G* and *Figure 7—figure supplement 1D*). Taken together, our studies provided compelling evidence that the transition that HF placodes naturally undergo from a columnar to conical shape is driven by the downregulation of RHOU and associated changes in actin dynamics. Moreover, our gain and loss of function experiments further demonstrate that this dampening of RHOU expression is essential for downgrowth of the follicle. Finally, these data imply that by accentuating the conical cell shape required for HF invagination, *shRhou* cells acquire a competitive advantage for invasion and HF downgrowth over wild-type cells in a mosaic tissue and are more likely to contribute to the HF (*Figure 7H*).

## Discussion

### An unprecedented morphogenetic screen in mice

Many of the pioneering genetic screens in *Drosophila* have focused on the epidermis and its appendages, where phenotypic abnormalities can be readily spotted and analyzed. By screening for enrichment or depletion of shRNAs in the epidermis and HFs, we've illustrated the power of our strategy in identifying novel regulators of mammalian skin morphogenesis. The methodology we've developed is broadly applicable, as the single E9.5 progenitor layer also gives rise to mammary glands, sweat glands and oral epithelia. Moreover, since adjusting infection times can achieve transduction of different tissues, including neuroectoderm and internal tissues, morphogenetic screens using lentiviral transductions should be applicable to interrogate many processes across mammalian development.

Screening for Rho GTPases and their regulators in morphogenesis offered an interesting challenge because of their complexity and broad expression patterns. To date, only the original three Rho GTPase family members and a handful of their regulators have been studied in depth in vivo. The epidermis and HFs proved to be the perfect morphogenetic system to probe this complexity, given their diverse developmental repertoire of cytoskeletal-based changes in cell shapes, cell fate specifications, cell movements and intercellular and cell-basement membrane interactions needed to achieve these proper and remarkably distinct tissue architectures.

Taking an unbiased screening approach was the only way to illuminate non-redundant distinctions among different members of this superfamily. That said, we were surprised to find that 42% of the genes targeted by our library showed an enrichment or depletion of their targeting shRNA in either the epidermal or HF fractions, revealing considerably less redundancy than previously assumed for Rho family members and their regulators. When taken together with the fascinating diversity in phenotypes that we observed upon analyses of only a few of these hits, our screen exposes a treasure-trove of morphogenetic processes that are coupled to these critical regulators of cytoskeletal dynamics.

In reflecting on the many hits we encountered, *Rhob* and *Rac3* were surprising. Both *Rhob* (*Liu et al., 2001*) and *Rac3* knockout mice (*Corbetta et al., 2005*) are viable and fertile and don't show obvious defects during skin development. As we only scored embryonic phenotypes here, postnatal gene compensation by other Rho GTPases might explain these discrepancies. An alternative explanation comes from recent studies by Didier Stainier's lab, which have show that some mechanisms of gene compensation that result from gene knockouts are not triggered upon RNAi-mediated knockdown (*El-Brolosy et al., 2019*). Conversely, RNAis may fail to reveal gene function due to incomplete protein depletion. Ultimately, the combination of RNAi and gene knockout strategies may be necessary to illuminate the physiological relevance and full scope of a gene's function.

### RHOU, intercellular adhesion and planar cell polarity

Our screen identified not only prototypical Rho GTPases, but also atypical members like RHOU, as critical regulators in skin morphogenesis, and by coupling our screen results with developmental studies and proximity ligation proteomics, we gained further insights. Our realization that RHOU functions at adherens junctions was piqued by our prior knowledge that in developing mouse epidermis, both adherens junctions and cortical actin are required to polarize the CELSR1 cadherin and establish PCP guidance cues along anterior-posterior interfaces of the epidermis (*Devenport and Fuchs, 2008*; *Luxenburg et al., 2015*; *Devenport et al., 2011*; *Aw et al., 2016*).

Two new findings emerged from our subsequent functional analyses. First, we discovered that RHOU loss markedly perturbed the actin-driven formation of nascent adherens junctions, which arise during the narrow developmental window when planar polarization of CELSR1 takes place. Secondly, RHOU loss did not prevent the passive formation of adherens junctions that occurs in densely packed epithelial sheets. This explains why RHOU loss has not revealed intercellular adhesion defects in some contexts (*Loebel et al., 2011*), while in others, mislocalization of intercellular junction molecules has been described (*Dickover et al., 2014*).

Our discovery that RHOU acts upstream of PCP establishment also came with new insights. Initial links between PCP and Rho GTPases came from *Drosophila*, where parallels were found between PCP patterning phenotypes and hypomorphic RHOA mutations involving internal excisions in the

protein (**Strutt et al., 1997**). Similar mutants also disrupt the planar polarized ciliary beating that occurs in the mucociliary epidermal cells of Xenopus (**Park et al., 2008**). Although such dominant negative Rho mutations clearly disrupt PCP (**Eaton et al., 1996**), only triple/quadruple genetic mutants in *Drosophila* Rho GTPases have unveiled PCP phenotypes, and even those were mild (**Muñoz-Descalzo et al., 2007**).

Similar redundancy of Rho GTPases has been observed in other animals where PCP comes into play. Thus for instance, the worm RHOU orthologue, CHW-1, has also been implicated as an upstream GTPase involved in vulval development (**Kidd et al., 2015**). That said, drawing a link to PCP was only achieved with compound mutations involving other non-Rho pathway members (**Kidd et al., 2015**), as were vertebrate studies on other Rho GTPases associated with regions of convergence-extension (**Habas et al., 2001**; **Simões et al., 2014**). In the developing cochlea, a role for the two prototypical Rho GTPases RAC1 and CDC42 was shown to regulate auditory hair PCP (**Kirjavainen et al., 2015**; **Grimsley-Myers et al., 2009**).

Our findings now place newfound emphasis on mammalian RHOU as a key player in PCP establishment, and one that on its own, is sufficient to reveal strong PCP defects upon shRNA knockdown. As our studies further reveal, RHOU's function in governing PCP patterning in the skin is through the actin-mediated dynamics that drive adherens junction formation and dictate cell shape. These features constitute the overarching foundation of PCP establishment.

## RHOU as an integrator of the inverse relation between cell-cell and cell-substratum adhesion

In a number of different cell lines in vitro, RHOU has been implicated as a regulator of focal adhesion dynamics, with elevated RHOU leading to focal adhesion dissolution and reduced RHOU favoring more stable focal adhesions (**Chuang et al., 2007**; **Ory et al., 2007**; **Ruusala and Aspenström, 2008**). In this regard, our proteomics were particularly intriguing as they illuminated RHOU at the heart of the protein complex that can shuttle between and govern an inverse relation between focal adhesions and adherens junctions. In vivo depletion of members of this complex showed similarities to RHOU loss of function, with an increase in the basal (focal adhesion) surface of epidermal progenitors and a decrease in apico-lateral (adherens junction) surfaces. Conversely, our gain of RHOU studies were consistent with an increase in apico-lateral domains at the expense of basal domains. It will be interesting in the future to see the extent to which RHOU functions in other contexts, including malignancy and wound healing, where reductions in the actin dynamics associated with adherens junction formation are often counterbalanced by more motile protrusion dynamics associated with focal adhesions turnover.

Pursuing the role for RHOU in cell-cell junction dynamics, it was also notable that the top hit after the junction-shuttling proteins was p120-CATENIN (*Ctnnd1*). While previously not identified as a RHOU-interacting partner, p120-CATENIN is an established E-CADHERIN-binding protein which in polarized epithelia, exists in a complex with RhoGEFs/GAPs to alter associated Rho GTPases and affect apico-lateral junction stability (**Klompstra et al., 2015**; **Lang et al., 2014**; **Noren et al., 2000**; **Wildenberg et al., 2006**). By differentially expressing RHOU, known to have a higher intrinsic nucleotide exchange rate than CDC42, we posit that these junctional zones may lessen their reliance on other GTPase regulators, thereby acquiring the means to prolong stability of adherens junctions and columnar cell shape. Further support this tantalizing notion comes from studies on the foregut endoderm, where RHOU is expressed in columnar and absent in more squamous tissues (**Loebel et al., 2011**). Intriguingly, *Rhou* mRNA has recently been shown to be more highly expressed in the differentiated progeny versus the stem cells of the gut epithelium (**Slaymi et al., 2019**). Similarly to what we observed in the epidermis, inactivation of *Rhou* in gut increases significantly the level of pMLC2 and alters cell contractility. In contrast to the developing epidermis, however, depletion of RHOU in the gut leads to an increase in cell proliferation and a reduction of apoptosis, suggesting that RHOU signaling may be tailored to suit the particular needs of each tissue.

## Changes in RHOU expression elicit dynamic actin-mediated cell shape remodeling in morphogenesis

Our studies revealed that within the epidermal plane, assembly of a columnar epithelium, reinforced by RHOU-activated formation of a cortical actin band, favors adherens junction assembly and

polarization of CELSR1, which in turn will determine HF angling along the anterior-posterior axis (*Figure 8*). Emerging placode cells within the epidermal plane also begin by displaying an elongated shape. However, as HF morphogenesis proceeds, this cellular architecture soon wanes and the cells switch to a conical shape, displaying an accentuated basal and a constricted lateral-apical surface that facilitates invagination and hair bud downgrowth (*Figure 8*). This process requires Myosin IIA, and *Myh9* mutant hair buds often fail to invaginate (*Le et al., 2016*).

Our findings now place RHOU at the crux of these actomyosin and cell shape dynamics that typify HF morphogenesis. As our gain of function studies revealed, RHOU maintains epidermal architecture, but this antagonizes HF invagination and downgrowth. Thus for HF morphogenesis to occur and for hair buds to invaginate, RHOU expression must be downregulated. Since *Rhou* shRNAs were selected for in the HF fraction of our screen, our findings further illustrate that by accelerating the switch to a conical shape, downregulating RHOU stimulates the invagination process relative to RHOU-expressing cells. Our findings are interesting in light of prior studies showing that in early embryos derived from *shRhou*-transduced embryonic stem cells, the developing ventral foregut endoderm also fails to maintain its apical position within the epithelium (*Loebel et al., 2011*). Taken together, we speculate that RHOU suppression may be a general feature of morphogenetic processes that require invagination from a RHOU+ epithelium.

In summary, our morphogenesis screen and the plethora of new potential RHOU interactions highlight a role for RHOU in choreographing the architecture of the epidermis as well as the HF (*Figure 8*). As deeper genetic analyses are conducted and interactome associations are validated, further insights into this fascinating atypical Rho GTPase will emerge.

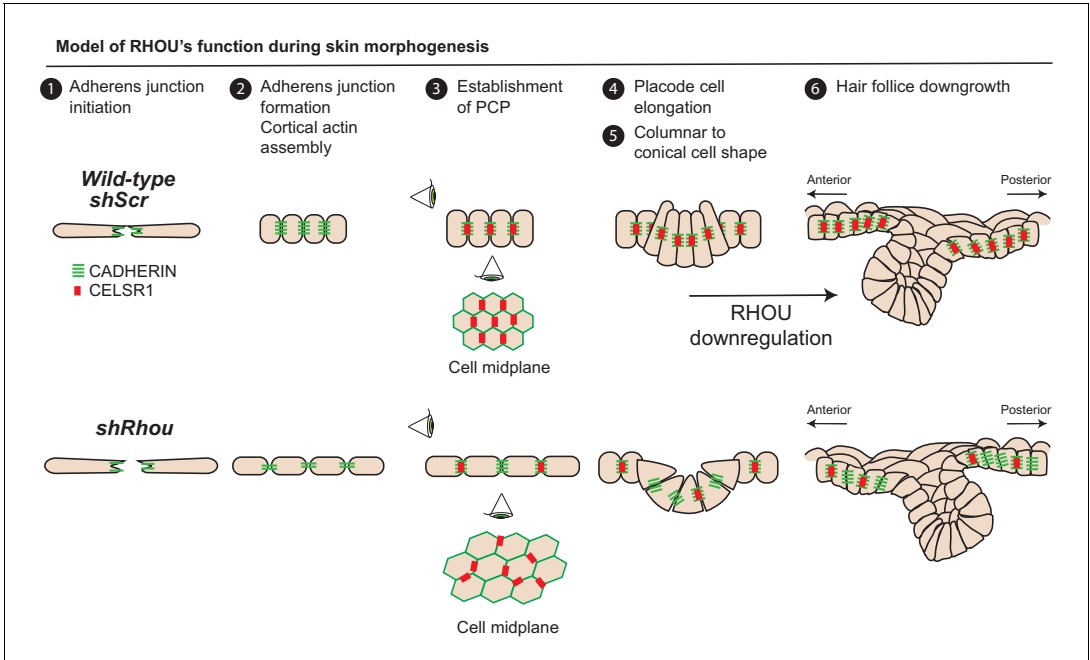

**Figure 8.** Model of RHOU's function during skin morphogenesis. RHOU organizes the cortical actin network and promote adherens junctions formation required to establish CELSR1 and PCP within the basal plane of the epidermis. PCP establishment within the epidermis is required to orient the angling of hair follicle downgrowth. RHOU also binds and orchestrates the localization of a shuttling complex that controls the inverse relation between intercellular adherens junctions and integrin-mediated cell substratum junctions. As such, downregulation of RHOU generates the conical shape (enhanced basal surface and decreased apico-lateral surfaces) required for HF invagination and downgrowth.
DOI: https://doi.org/10.7554/eLife.50226.025

## Materials and methods

### Animal models

The following mouse strains were used: CD1/ICR (Charles River Strain, code 022), *Krt14-Actin-GFP* (*Vaezi et al., 2002*), *Krt14-rTTa* (*Nguyen et al., 2009*). All mouse strains were housed in an AAA-LAC-accredited facility and experiments were conducted according to the Rockefeller University's Institutional Animal Care and Use Committee, and NIH guidelines for Animal Care and Use. All animal procedures used in this study are described in our #17020-H protocol named *Development and Differentiation in the Skin*, which had been previously reviewed and approved by the Rockefeller University Institutional Animal Care and Use Committee (IACUC). For in utero injections, pregnant females were used at E9.5. Both male and female embryos were injected with lentivirus in utero and collected at the indicated embryonic time (E13.5, E14.5, E15.5, E16.5 and E18.5) before their analysis. For MYC-RHOU expression, *Krt14-rtTA* was activated by feeding pregnant females with doxycycline (2 mg/kg, Doxyfeed, Bio-Serv) chow at E9.5 until time of collection.

### Primary cell culture

Primary mouse keratinocytes were purified as previously described (*Blanpain et al., 2004*). Briefly, epidermal keratinocytes were isolated and cultured from dispase-treated skins of P0 wild-type CD1 or *Krt14-rtTA* pups. Cells were cultured in 37˚C incubator in the presence of 7.5% $CO_2$. Cells were passaged and maintained in E media with low calcium (0.05 mM $CaCl_2$). Sex of mouse primary keratinocytes is not available.

### Lentiviral library and RNAi screen

RNAi screen and gene knockdowns were performed in CD1 mice. An shRNA-pooled library of 999 shRNAs obtained from the Broad Institute Mission TRC-1 and TRC-1.5 libraries (Sigma) targeting 166 individual genes was generated. Large-scale production and concentration of lentivirus were performed as previously described (*Beronja et al., 2010*). For t0 in vitro samples, two 10 cm dishes of confluent epidermal keratinocytes previously purified from CD1 pups were infected with 1 µl of titrated lentiviral library and collected 24 hr after transduction. In utero lentiviral injection was performed at E9.5 using 1 µl of titrated library to selectively transduce the surface ectoderm of mouse embryos at a MOI $\leq$ 1. The RNAi screen was performed in triplicate and the material from 60 embryos was pooled per biological replicates. Based on library size and the amount of HF progenitors, each biological replicate had a coverage of $\geq$100 fold.

The skin of E18.5 embryos was removed and put in dispase (Sigma, T3934) for 1 hr at 37˚C, the epidermal and dermal fractions were separated and processed separately in a 1:1 mixture of 0.25% Trypsin-EDTA (Gibco, 25200056) and Versene (Gibco, 13151014) for 20 min. Single cell suspension was achieved by filtering through a 40 µm cell strainer. Primary keratinocytes and epidermal cells from epidermal and HF fractions were used for genomic DNA isolation using the DNeasy Blood and Tissue Kit (Qiagen, 69504). The genomic DNA from 60 transduced embryos was pooled, and 10% of the total material for each biological sample (19 µg of gDNA for the epidermal fraction and 291 µg of gDNA for the HF fraction) were used as template for a 28 cycles PCR reaction using Phusion High-Fidelity DNA Polymerase (NEB, M0530) and according to manufacturer guidelines. The use of distinct custom primers for each biological sample PCR reactions allowed barcoding of samples and replicates (see *Supplementary file 2* for primer sequences). For our previous screens we had validated our barcodes, and documented that our pre-amplification and sequencing reactions did not bias shRNA quantifications (*Beronja et al., 2013*). PCR products were precipitated, cleaned using DNA Clean and Concentrator-5 column (Zymo Research, D4031) and run on a 2% gel. A clean ~200 bp band was isolated using QIAquick Gel Extraction Kit (Qiagen, 28704). Samples for each condition and replicate were sent for Illumina NextSeq500 sequencing.

Illumina reads were trimmed to hairpin sequences (21 nucleotides) and aligned to library. The shRNAs showing zero reads were discarded from the analysis. Hits were classified based on a number of criteria: 1- more than two shRNAs for a gene showed an absolute two fold change in their representation. 2- the fold change has an adjusted *P* value < 0.05 using multiple two-tailed t-test and the original false discovery rate (FDR) method of Benjamini and Hochberch, with a FDR set at 10% (*Asare et al., 2017*; *Sendoel et al., 2017*). 3- No shRNA targeting the same gene showed

opposing trends. Importantly, because each E9.5 progenitor generates ~40 cells by E18.5 ($1.2 \times 10^7$ epidermal cells and $6.8 \times 10^6$ HF cells total), growth-neutral shRNAs should be represented ~4,000 fold in the E18.5 sequencing reaction, ensuring robustness for our screen (*Beronja et al., 2013*).

## Constructs and shRNAs

All shRNA TRC library clones used in library can be found in *Supplementary file 1*. The hairpin sequences used for candidates studies where: *Fgd2* (TRCN0000110041): 5'-CTGATGTCTATACCGAGACAA-3'; *Fgd2* (TRCN0000110044): 5'-TGATGTCTATACCGAGACAAT-3'; *Trio* (TRCN0000254106): 5'-ACGTACACAAACGCAGATAAA-3'; *Trio* (TRCN0000254107): 5'-TCGACCTA TCCGTAGCATTAA-3'; *Rhou* (TRCN0000077504): 5'-GCTACAGCCAAAGAAGTCTAA-3'; *Rhou* (TRCN0000077505): 5'-GACGTCAAAGTGCTCATAGAA-3'; *Myh9* (TRCN0000071504): 5'-CGGTAAA TTCATTCGTATCAA-3'; *Ctnnb1* (TRCN0000012690): 5'-GCTGATATTGACGGGCAGTAT-3'; *Pak2* (TRCN0000025209) 5'-CGATGAAGAGATTATGGAGAA-3'; *Arhgef7* (TRCN0000324966): 5'-CAGA TCCTGAAGGTTATCGAA-3'. For in vivo studies shRNAs were digested using SacII/Sph1 from the *pLKO* library clone, purified and inserted in a *LV-pLKO-U6-stuffer-pgk-H2B-RFP* (*Beronja et al., 2010*) vector using SphI and SacII. The resulting clones were used for in utero injections. Importantly, the two shRNAs targeting *Rhou* do not share significant homology with the close member *Rhov* and differential gene expression analysis revealed that *Rhou* is the main Rho GTPases differentially expressed in *shRhou*-transduced cells. To generate *Rhou* expression construct, mouse *Rhou* cDNA was PCR amplified using a *Myc* containing primer from *pCMV6-Rhou* (Origene, MR219421) and inserted into a *LV-Tre-pgk-H2B-RFP* construct (*Chang et al., 2013*). To generate the *LV-Myc-BioID2-GFP-pgk-H2B-RFP* doxycycline inducible expression construct, the *Myc-BioID2* cDNA and the *GFP* cDNA were PCR amplified from *Myc-BioID2-pBABE-puro* (Addgene, 80900) and *LV-U6-stuffer-pgk-H2B-GFP* (*Beronja et al., 2010*) respectively and inserted into *LV-Tre-pgk-H2B-RFP construct*. To generate the LV-*Myc-BioID2-Rhou-pgk-H2B-RFP* expression construct, the *Myc-BioID2* cDNA and the *Rhou* cDNAs were PCR amplified from *Myc-BioID2-pBABE-puro* and *pCMV6-Rhou* respectively and inserted into *LV-Tre-pgk-H2B-RFP*.

## RNA isolation, quantitative PCR and RNA-sequencing

Primary keratinocytes and E14.5 epidermal cells (see below for sorting strategy) were lysed using TRI Reagent (Sigma, T3934) and RNA extraction was performed using Direct-zol RNA mini prep kit (Zymo Research, R2050) and by following manufacturer instruction. For qPCR analysis, equal amounts of primary keratinocytes RNA were used for cDNA synthesis using SuperScript VILO (Invitrogen, 11754050). Semi-quantitative PCR was done using the indicated primers below and the SYBR Green PCR Master Mix (Applied Biosystems, 4367659). Reactions were run for 50 cycles and followed by a melting curve analysis to establish specificity of the reaction. *Hprt* was used as a gene of reference. The sequences of the primers used were: *Trio*: Forward 5'-CAGTCCGTGCAGTCCAC TA-3', Reverse 5'-GCATGGTGGAGATGTTGCTA-3'; *Fgd2*: Forward 5'-TGAGAAGCGGAGTGAGACCT-3', Reverse 5'-GCTCCTCTACCTGTGGCTTG-3'; *Rhou*: Forward 5'-ACGGCCTTCGACAAC TTCT-3', Reverse 5'-ACTCATCCTGTCCTGCAGTGT-3'. *Hprt*: Forward 5'-GATCAGTCAACGGGGGACATAAA-3', Reverse 5'-CTTGCGCTCATCTTAGGCTTTGT-3'. *Arhgef7:* Forward 5'-CC TTGGAGCCTGATTGC-3', Reverse 5'-TGGTCTTGGGGCTCTTACTG-3'. For RNA-Sequencing analysis, total RNA was submitted to the Genomics Resources Core Facility of the Weill Cornell Medical College for quality control that was determined using Agilent 2100 Bioanalyzer. Library preparation was done using IlluminaTruSeq Stranded mRNA Sample Kits. Sequencing was achieved on an Illumina HiSeq4000.

## Flow cytometry and fluorescence-activated cell sorting

Purification of basal epidermal cells from CD1 E14.5 embryos transduced with *shScr* or *shRhou-505* was achieved by preparing single cell suspension as described above and was performed on a FACS Vantage SE system supplied with FACS DiVa software (BD Biosciences). Cells were gated for viability, singlet and sorted according to their negative expression of lineage markers (CD31⁻, CD45⁻, CD117⁻, CD140a⁻), and their positive expression of α6 integrin and RFP which marked the transduced cells. The following antibodies were used: CD49f-PE/Cy7 (1:1000, BioLegend, 313622), Biotin-CD31 (1:200, BioLegend, 102504), Biotin-CD45 (1:200, BD, 553077), Biotin-CD117 (1:200,

BioLegend, 105804), Biotin-CD140a (1:200, BioLegend, 135910), Streptavidin-FITC (1:500, BioLegend, 405202) and Streptavidin-APC/Cy7 (1:500, BD, 554063).

## Cell competition assay

For competition assay, *shScr-H2B-GFP* lentiviruses were mixed with *shScr-H2B-RFP*, *shMyh9-H2B-RFP*, *shCtnnb1-H2B-RFP* and *shRhou-H2B-RFP* lentiviruses in a 1:1 proportion. In utero lentiviral injection was performed at E9.5 using 1 µl of titrated mixes to selectively transduce the surface ectoderm of mouse embryos at a MOI $\leq$ 1. Litters were collected at E18.5, the skin was dissected, the epidermis was separated from the dermis fraction as previously described and single cells suspension was achieved as described above. The ratio of GFP$^+$ and RFP$^+$ cells was measured by flow cytometry, and the data were analyzed using FlowJo software (BD Biosciences).

## Embryo preparation, immunofluorescence, microscopy and image processing

To limit variability between mice, only animals with $\geq$80% transduction efficiency were used for phenotype analysis of candidate-depleted skin. For sagittal sections, embryos were embedded and frozen in OCT compound Tissue Tek (VWR, 25608–930). 12 µm cryosections were fixed for 10 min in 4% paraformaldehyde (PFA, From 16% PFA solution Electron Microscopy Sciences, 15700), washed in PBS, permeabilized for 10 min in 0.3% Triton, blocked for 1 hr in gelatin block (2.5% fish gelatin, 1% BSA, 2.5% normal donkey serum, 0.3% Triton in 1x PBS). Sections were incubated with primary antibodies diluted in gelatin block overnight, washed 3 × 5 min with PBS + 0.02% Tween and stained for 1 hr with secondary antibodies in blocking buffer. For whole-mount analysis, embryos were fixed for 1 hr in 4% PFA and washed overnight in PBS. Dissected head and back skin pieces were blocked for 2–4 hr in 0.3% Triton, incubated with primary antibodies diluted in blocking buffer overnight to two days, washed 3 × 30 min with PBS + 0.3% Triton, incubated with secondary antibodies diluted in blocking buffer overnight and washed 3 × 30 min with PBS + 0.3% Triton. Cryosections and whole-mounts were mounted with ProLong Gold antifade reagent with DAPI (Thermo Fisher, Life technologies 36935). For in vitro cell spreading and cell differentiation assay, cells were fixed for 10 min in 4% PFA and proceed similarly than sagittal sections. Images of cryosections and cultured cells were captured using an Axio Observer.Z1 inverted microscope (Zeiss) equipped with an ApoTome.2 (Zeiss), a Hamamatsu ORCA-ER camera (Hamamatsu Photonics) and either a 20x air objective, 40x oil immersion objective or 60x oil immersion objective. Image acquisition was controlled by Zen pro software (Zeiss). Images of whole-mounts were captured using an inverted LSM 780 laser scanning confocal microscope (Zeiss) and either a 20x air objective (NA = 0.8), or a 63x oil immersion objective (NA = 1.4). Basic image adjustments were performed in Fiji (ImageJ).

The following antibodies were used: ARHGEF7 (rabbit, 1:200, Millipore 07–2101), CELSR1 (guinea pig, 1:250, Fuchs Lab), E-CADHERIN (rat, 1:50, gift from Dr Takeichi), GFP (chicken, 1:2000, Abcam ab13970), Keratin 5 (guinea pig, 1:500, Fuchs Lab), Keratin 6 (guinea pig, 1:1000, Fuchs Lab), Keratin 10 (rabbit, 1:500, Covance PRB-159P), LORICRIN (rabbit, 1:2000, Covance PRB-145P), MYC (mouse, 1:200, Cell Signaling 2278), NCAM (rabbit, 1:200, Millipore AB5032), pMLC2 (rabbit, 1:200, Cell Signaling 3674), PAK2 (rabbit, 1:200, Cell Signaling 2608), P-CADHERIN (goat, 1:400, R and D AF761), RHOU (rabbit, 1:1000, OriGene TA344077), RFP (rat, 1:1000, Chromotek 5f8), VINCULIN (mouse, 1:200, Sigma V9131). Secondary antibodies conjugated to Alexa Fluor 488/546/647 were diluted 1:1000 for sagittal sections and 1:200 for whole-mounts. F-ACTIN was detected using Phalloidin 488/Rhodamin/647 (1:200, Life technologies).

## Cell transfection, cell spreading and cell differentiation assay

Mouse primary keratinocytes form CD1 pups were isolated and maintained as previously described (*Blanpain et al., 2004*). For transfection of shRNAs, $1 \times 10^5$ cells were plated in 12 wells plate and transfected the next day using 500 µg of *pLKO shRNA* containing vector and Effectene Reagent (Qiagen, 301425) and by using manufacturer guidelines. 48 hr after transfection, cells were selected with 1 ug/ml puromycin (ThermoFisher, A1113803) for 3 days. For transfection of MYC-RHOU, a confluent 10 cm plate was transfected using 4 µg of plasmid and Effectene Reagent (Quiagen, 301425) and by using manufacturer guidelines. For qPCR analysis, cells were lysed in TRI reagent (Sigma T3934) directly in plates and proceed for assay as described above. For cell spreading and cell

differentiation assay, Millicell EZ SLIDE four well glass slide (Millipore, PEZGSO416) were coated with 10 µg/ml human plasma fibronectin purified protein (Millipore, FC010) for 1 hr. For cell spreading assay, transfected and puromycin-selected keratinocytes for each condition were trypsinized, and $1 \times 10^4$ cells were plated in individual chambers and allowed to spread for 24 hr. Cells were then fixed and proceed to immunohistochemistry as described above. For cell differentiation assay, transfected and puromycin-selected keratinocytes for each condition were trypsinized and $1 \times 10^5$ cells were plated in individual chambers and allowed to spread and reached confluence before cells were put in the presence of 1.5 µM CaCl$_2$ for 2 hr. Cells were then fixed and processed for immunohistochemistry as described above. Focal adhesion characteristics were quantified using Fiji (ImageJ). Morphometric analysis was performed on thresholded images to select and classified objects of a size range of $0.3 \leq \infty$ as focal adhesion based on images of VINCULIN staining.

## Identification of RHOU's interactome using a BioID strategy and mass spectrometry

For the identification of RHOU's interactome, *K14rtTA* keratinocytes were infected with *LV-Tre-Myc-BioID2-GFP-H2B-RFP* or *LV-Tre-Myc-BioID2-Rhou-H2B-RFP* (described above). RFP$^+$ transduced cells were isolated by FACS for both population and stable cell lines were established. For the identification of RHOU's interactome during keratinocytes growth, two 15 cm culture dishes per conditions (*Myc-BioID2-Rhou* and *Myc-BioID2-GFP*) were seeded. To induce expression of recombinant proteins, 1 µg/ml doxycycline (Sigma, D9891) was added to growth media. Cells were allowed to grow for 4 days and were pulsed with 50 µM biotin (Sigma, B4501) in the presence of doxycycline for 16 hr before reaching confluence. For purification cells were treated as previously described (*Kim et al., 2016*; *Roux et al., 2012*). Briefly, cells were lysed in buffer (50 mM Tris pH 7.4, 500 mM NaCl, 0.2% SDS and 1 mM DTT supplemented with 1x cOmplete Protease Inhibitor Cocktail (Sigma, 11697498001) for 10 min on ice and 20% Triton was added to samples to reach a final concentration of 2%. Samples were sonicated and filtered on pre-washed Zeba Spin Desalting Columns (7K MWCO for 10 ml, ThermoFisher, 89893) according to manufacturer guidelines. Samples were diluted 1:1 in 50 mM Tris pH 7.4 and centrifugated at 16,000 g for 10 min. 250 µl of pre-washed Dynabeads MyOne Streptavidin T1 (ThermoFisher, 65601) were added to the cleared samples and leave to incubate overnight at 4°C. Samples were incubated with DynaMag-2 Magnet (ThermoFisher, 12321D), washed 1x in buffer 1 (2% SDS), washed 2x with buffer 2 (0.1% deoxycholic acid, 1% Triton, 1 mM EDTA, 500 mM NaCl and 50 mM HEPES pH 7.5), washed 2x with buffer 3 (0.5% deoxycholic acid, 0.5% NP-40, 1 mM EDTA, 250 mM LiCl, 10 mM Tris pH 7.4), washed 3x with 2 M urea and washed 2x with PBS. All washes were performed using a magnetic stand. New tubes were used in between each urea and PBS washes. Samples were re-suspended in 500 µl of 50 mM Tris pH 8.0. Wash buffer was removed from suspension of magnetic beads and replaced with 100 µl 8M Urea, 50 mM ammonium bicarbonate, 10 mM dithiothreitol. After 1 hr the solution was removed and replaced with 100 ul 40 mM iodoacetamide and incubated in the dark for 30 min. Alkylation solution was replaced with 1 µg trypsin (Promega) dissolved in 100 µl 50 mM ammonium bicarbonate and incubated for 4 hr. Supernatant was then removed and re-digested overnight using 0.5 µg trypsin and 0.5 µg Endopeptidase Lys-C (Wako). Peptides were desalted and concentrated using C18 based Stage tips (*Rappsilber et al., 2007*) and separated by nanoLC (gradient: 2%B/98%A to 38%B/62%A in 70 min, A: 0.1% Formic Acid, B: 90% acetonitrile/0.1% Formic acid) coupled to a Fusion Lumos operated in high/high mode. Data were queried against UniProts Complete Proteome mouse database (March 2016, 51290 sequences) concatenated with common contaminants. ProteomeDiscoverer v. 1.4.0.288 (Thermo Scientific) combined with Mascot v. 2.5.1 (Matrix Science) was used for the analysis. Results were filtered using a Percolator (*Käll et al., 2007*) calculated peptide False Discovery Rate of 1%. Proteins were considered part of RHOU's proximity interactome if they were identified in 2 of the 3 MYC-BioID2-RHOU replicates and absent of all of the MYC-BioID2-GFP samples.

## Western blotting

Proteins were extracted in RIPA lysis and extraction buffer (ThermoFisher, 89900) containing protease inhibitors (cOmplete Protease inhibitor Cocktail, Sigma, 11697498001) and phosphatase inhibitors (phosSTOP, Sigma 4906845001). For co-Immunoprecipitation protein were extracted in Pierce IP Lysis Buffer (ThermoFisher Scientific, 87788) containing protease and phosphatase inhibitors. A

total of 500 µg of protein was incubated overnight with anti-MYC antibody. The next day 100 µl of washed Pierce Protein A/G agarose beads slurry (ThermoFisher Scientific, 20421) was added for an additional 2 hr. Beads were wash 3x with lysis buffer and proteins were eluted in SDS sample buffer. All samples were run on a NuPAGE 4–12% Bis-Tris Protein Gels (ThermoFisher, NP0321), transferred on PVDF membrane. Membranes were blotted in TBS-0.1% tween + 1% bovine serum albumin overnight with primary antibodies, washed $3 \times 5$ min with TBS-0.1% Tween, incubated with secondary antibody for 30 min, washed $3 \times 5$ min with TBS-0.1% Tween and reveal using Clarity Western ECL Substrate (BioRad, 179–5060). The following antibodies were used: α-TUBULIN (mouse, 1:10,000, Sigma T5168), RHOU (rabbit, 1:1000, OriGene TA344077), MYC (mouse, 1:1000, Cell Signaling 2278), GFP (chicken, 1:10 000, Abcam ab13870), pPAK1$^{Ser144}$/pPAK2$^{Ser141}$ (rabbit, 1:1000, Cell Signaling 2606), PAK1 (rabbit, 1:1000, Cell signaling 2602), PAK2 (rabbit, 1:1000, Cell signaling 2608), PAK1/2/3 (rabbit, 1:1000, Cell signaling 2604) pMLC2$^{Thr18/Ser19}$ (rabbit, 1:1000, Cell Signaling 3674) and MLC2 (rabbit, 1:1000, Cell Signaling 3672), ARHGEF7 (rabbit, 1:1000, Millipore 07–2101).

## Quantification of planar cell polarized cells and hair follicle orientation analysis

To assess HF orientation, whole-mounts samples from head skin of E18.5 embryos were stained with P-CADHERIN and RFP and low magnification Z-stacks were acquired. HF were classified as either perpendicular to the basal plane or angled. The orientation of the angled HF according to the anterior-posterior plane was determined by drawing a line between the base and the tips of the hair using the Fiji straight-line tool. An angularity of 0° represent a perfect alignment of the HF along the anterior-posterior axis of the embryo. To assess the establishment of PCP in the epidermis, whole-mount samples from head skin of E14.5, E15.5 and E16.5 embryos were stained with CELSR1 and E-CADHERIN and imaged as described above. Data collection was conducted using confocal image at a horizontal plane from the middle of the basal cell layer. A PCP polarized basal cells was defined as a cell in which two opposing domains of CELSR1 can be observed. The angle of polarity was determined using the Fiji straight-line tool.

## Quantitative analysis of P-CADHERIN and F-ACTIN pixel intensity

P-CADHERIN and F-ACTIN pixel intensity was measured using Fiji (ImageJ) straight line tool and by using the function Plot Profile. Data collection was conducted using confocal images at a horizontal plane from the middle of the basal cell layer from E14.5 and E15.5 embryos stained with P-CADHERIN and F-ACTIN. A straight line across individual cell was drawn and the pixel intensity profile plotted. Pixel intensity across individual cell was normalized to account for differences in cell size by dividing each cell in 20 equal bins.

## Quantitative analysis of basal and placode cell shape dynamics, cell density and number of cell in placode

Measurements of basal cell area, placode cell area, cell eccentricity and cell density were performed using Cell Profiler (*Jones et al., 2008*; *Lamprecht et al., 2007*) and their own tissue neighbors pipeline (www.cellprofiler.org). Briefly confocal images of basal cell in the epidermis were segmented on the basis of E-CADHERIN to measure basal cell area. Z-stack confocal images from placodes were acquired and orthogonal views were used to identify the apex, middle and bottom planes that were than segmented on the basis of P-CADHERIN to measure placode cells area. The number of placode cells in the middle plane was recorded. Basal cell height was measured using the Fiji straight-line tool on confocal images from E15.5 and E16.5 embryos stained with Keratin 5. Placode area was determined using confocal images and the Fiji area tool.

## Quantification of cell proliferation

For cell proliferation assay, pregnant female mice were injected with 5-ethynyl-2'deoxyuridine (EDU) intraperitoneally allowing E18.5 embryos transduced with *Scr or Rhou-505* shRNAs to be pulsed for 3 hr. Embryos were dissected, frozen in OCT, sectioned (12 µm) and processed according to the manufacturer's instruction (Click-iT EDU Alexa Fulor 647 Imaging kit, Life Technologies, C10340). The ratio of RFP$^+$ cells (transduced cells) that are EDU$^+$ was calculated for each condition.

## Barrier assay

Dye penetration assay was performed as previously described (*Asare et al., 2017*). Briefly, E17.5 and E18.5 embryos were isolated form the pregnant mother. Euthanized embryos were immersed in ice cold PBS for 30 min. Embryos were immersed in cold methanol gradient in water, taking 2 min per step (1–25%, 2–50%, 3–75%, 4–100% methanol) and rehydrated in methanol gradient in water, taking 2 min per step (1–75% methanol, 2–50% methanol, 3–25% methanol, 4–100%). Embryos were immersed in 0.1% toluidine blue solution in water on ice. Embryos were destained in PBS to reveal dye pattern and barrier properties.

## RNA-Seq alignment and differential expression analysis

Embryos from the same litters were pooled and used as individual biological replicates. Total RNA was isolated from FACS purified basal epidermal cells, (see above for a description of the purification strategy) using Direct-zol RNA MiniPrep kit (Zymo Research) per manufacturers instruction. RNA quality was determined using Agilent 21100 Bioanalyzer and all samples had a RIN >0.8. Poly-A enrichment and library preparation using Illumina TruSeq mRNA sample preparation kit was performed by Weill Cornell Medical College Genomic Core Facility. Samples were sequenced on Illumina HiSeq4000 and 50 bp Single-end reads obtained. Transcripts were quantified to generate TPM and counts using Salmon (*Patro et al., 2017*) with Gencode M18 (mm10) used as the reference. Differentially expressed genes were determined using DESeq2 (*Love et al., 2014*) with lfcThreshold = 0 and alpha = 0.01 contrasting *shRhou* and *shScr*.

## Statistic and blinding

Statistical analyses were performed using Prism 8 (Graphpad) software and representative data are shown. All experiments were repeated at least three times. For each measurement, at least three biological replicates were used and littermates used as controls. No statistical method was used to predetermined sample size, randomization and experiment blinding was not used. Quantitative data were represented as the mean and standard error of the mean (SEM). Normal distribution of the data was determined using the Shapiro Wilk test. Quantitative data that followed a normal distribution were compared using either a paired or unpaired parametric two-tailed *t*-test. Quantitative data that didn't meet the criteria for a normal distribution were compared using a non-parametric two-tailed Mann-Whitney test. Significance of *P* value was set at <0.05. Individual *P* values are provided in the figures. Correlation between samples was determined using Spearman correlation. Changes in distribution frequency were determined using Kolmogorov-Smirnov test. Statistical details for each experiment, including the statistical test used, the sample size for each experiment, the *P* and $R^2$ value can be found in the corresponding figure legend.

## Acknowledgements

We thank the Rockefeller University's Comparative Biology Research Center, an AAALAC-accredited animal facility, for their assistance in animal husbandry and housing for all of our mouse studies. We also thank L Polak and L Hidalgo for their assistance with some of the mouse experiments performed in this study. We thank additional Rockefeller Resource Centers, without which this work would not be possible: Flow Cytometry Facility (S Maizel, director), Bioimaging Facility (A North, director), Proteomics Facility (H Molina, Director). The Rockefeller University Proteomics Resource Center acknowledges funding from the Leona M and Harry B Helmsley Charitable Trust and Sohn Conferences Foundation for mass spectrometer instrumentation. Our RNA Sequencing was performed through the facility at Cornell Weil School of Medicine. We are grateful to the entire Fuchs' lab, but especially S Ellis, Y Ge, A Asare and H Yang for providing valuable feedback and I Matos for microscopy expertise. EF is a Howard Hughes Medical Institute investigator. ML was the recipient of a Canadian Institutes of Health Research postdoctoral fellowship and currently holds a Scholarship for the Next Generation of Scientists from the Cancer Research Society. NCG holds a Postdoctoral Enrichment Program Award from the Burroughs Wellcome Fund and is supported by a NIH Postdoctoral Ruth L Kirschstein National Research Service Award F32CA221353. AS was the recipient of sequential postdoctoral fellowships from the Human Frontiers Science Program and the Marie Curie

Foundation. The work was supported by grants from the National Institutes of Health (R01-AR27883, EF).

## Additional information

### Competing interests
Elaine Fuchs: Reviewing editor, *eLife*. The other authors declare that no competing interests exist.

### Funding

| Funder | Grant reference number | Author |
|---|---|---|
| Howard Hughes Medical Institute | | Elaine Fuchs |
| Canadian Institutes of Health Research | Postdoctoral fellowship | Melanie Laurin |
| Burroughs Wellcome Fund | Postdoctoral Enrichment Program Award | Nicholas C Gomez |
| National Institutes of Health | Ruth L Kirschstein National Research Service Award (F32CA221353) | Nicholas C Gomez |
| Human Frontier Science Program | Postdoctoral fellowships | Ataman Sendoel |
| Marie Curie Foundation | Postdoctoral fellowships | Ataman Sendoel |
| Cancer Research Society | Scholarship forthe Next Generation of Scientists | Melanie Laurin |
| National Institutes of Health | R01-AR27883 | Elaine Fuchs |

The funders had no role in study design, data collection and interpretation, or the decision to submit the work for publication.

### Author contributions
Melanie Laurin, Conceptualization, Data curation, Formal analysis, Validation, Investigation, Methodology, Writing—original draft; Nicholas C Gomez, Conceptualization, Formal analysis, Methodology; John Levorse, Megan Sribour, Methodology; Ataman Sendoel, Formal analysis; Elaine Fuchs, Conceptualization, Supervision, Funding acquisition, Writing—original draft, Project administration

### Author ORCIDs
Elaine Fuchs https://orcid.org/0000-0002-7198-3257

### Ethics
Animal experimentation: All mouse strains were housed in an AAALAC-accredited facility and experiments were conducted according to the Rockefeller University's Institutional Animal Care and Use Committee, and NIH guidelines for Animal Care and Use.All animal procedures used in this study are described in our #17020-H protocol named *Development and Differentiation in Skin,* which had been previously reviewed and approved by the Rockefeller University Institutional Animal Care and Use Committee (IACUC).

### Decision letter and Author response
Decision letter https://doi.org/10.7554/eLife.50226.037
Author response https://doi.org/10.7554/eLife.50226.038

## Additional files

### Supplementary files

• Supplementary file 1. shRNA Library Composition.
DOI: https://doi.org/10.7554/eLife.50226.026

• Supplementary file 2. Sequence Based Reagents.
DOI: https://doi.org/10.7554/eLife.50226.027

• Supplementary file 3. Genes With ≥Two shRNAs Showing an Absolute Enrichment or Depletion in The Hair Follicle Fraction.
DOI: https://doi.org/10.7554/eLife.50226.028

• Supplementary file 4. Genes With ≥Two shRNAs Showing an Absolute Enrichment or Depletion in The Epidermal Fraction.
DOI: https://doi.org/10.7554/eLife.50226.029

• Supplementary file 5. Genes With ≥Two shRNAs Showing an Absolute Enrichment or Depletion Only in The HF Fraction.
DOI: https://doi.org/10.7554/eLife.50226.030

• Supplementary file 6. List of RHOU's Interaction Partner in Growth Conditions.
DOI: https://doi.org/10.7554/eLife.50226.031

• Supplementary file 7. Key Resources Table.
DOI: https://doi.org/10.7554/eLife.50226.032

• Transparent reporting form
DOI: https://doi.org/10.7554/eLife.50226.033

### Data availability

Sequencing data have been deposited in NCBI GEO under accession number GSE123047. All data generated or analysed during this study are included in the manuscript and supporting files.

The following dataset was generated:

| Author(s) | Year | Dataset title | Dataset URL | Database and Identifier |
|---|---|---|---|---|
| Laurin M, Gomez NC, Levorse J, Sendoel A, Sribour M, Fuchs E | 2019 | RNA-sequencing from E14.5 epidermal cells from shScr and shRhou transduced mice | https://www.ncbi.nlm.nih.gov/geo/query/acc.cgi?acc=GSE123047 | NCBI Gene Expression Omnibus, GSE123047 |

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
