## [Decision Letter]

Thank you for submitting your article "An RNAi Screen Unravels The Complexities of Rho GTPase Networks in Skin Morphogenesis" for consideration by *eLife*. Your article has been reviewed by three peer reviewers, and the evaluation has been overseen by a Reviewing Editor and Anna Akhmanova as the Senior Editor. The reviewers have opted to remain anonymous.

The reviewers have discussed the reviews with one another and the Reviewing Editor has drafted this decision to help you prepare a revised submission.

This study employs an in vivo shRNA pool screen on embryonic epidermis and hair follicles to identify Rho GTPase effectors that regulate skin development. After validating the approach, a customized pool of selected shRNAs targeting Rho GTPases and their regulators were selected based on the enrichment or depletion of a given shRNA over time in the developing skin indicating positive or negative selection. 3 of these hits were followed up, with *RhoU* being analysed in some detail. The authors show that depletion of *RhoU* leads to a planar cell polarity defect as well as alterations in cell shape both in the epidermis and hair follicle. The cellular mechanism appears to in part involve increased cell matrix adhesion and actin anchored at these sites, concomitant with delayed formation of cell-cell contacts and organization of cortical actin. The findings are interesting and provide new insight into actin regulation and skin biology. A number of points were identified that require addressing:

Essential revisions:

1) The authors state that they used the 'most potent' shRNA of those present in the screen – but all three *RhoU* shRNAs tested were equally potent at reducing *Rhou* mRNA levels, thus the text needs altering here. Other *RhoU*, Trio and Fgd2 shRNAs replicate the initial hair follicle findings. This was done for two read-outs in Figure 4 only for *RhoU*, but not for other critical experiments that the authors spend most of the rest of the manuscript investigating i.e. effects on epidermal/keratinocyte cell shape, actin cytoskeleton and cell-cell junctions. These results need to be repeated with another *RhoU* shRNA to prove that they are not off-target effects.

2) Figure 5B: what is the variability between experimental results between mice? Is each cell an individual variable in the statistical analysis? The authors should rather consider each experiment (mouse) as an independent entity for comparison, because inevitably the knockdown will vary per mouse.

3) Figure 6: Presumably PAK2 rather than PAK1 was the strongest hit in the *RhoU* interactome because it is more highly expressed than PAK1 (or PAK3) in the epidermal cells used for the screen. This needs to be verified. To further support the possible involvement of PAK, activation of PAK1 and PAK2 should be tested in *RhoU* depleted keratinocytes by western blotting for phosphorylated forms of the kinases.

4) The cell shape changes as well as the changes in focal adhesions could all be explained by altered contractility, which has also been shown to be downstream of PAK. The authors could consider staining or western blotting for phosphorylated myosin light chain to strengthen the mechanistic aspect of the manuscript.

5) The screen identified *Rac1* as a regulator of hair follicle morphogenesis, which is supported by hair formation defects in knockout mice. However, it also identified *RhoB* and *Rac3*, where corresponding knockout mice do not show obvious hair follicle defects. *Cdc42* was surprisingly not detected as an important regulator of hair follicle formation, although it is known that *Cdc42* knockout in keratinocytes has a significant effect on hair follicle maintenance. These discrepancies must be discussed in the context of the field.

---

## [Author Response]

Essential revisions:1) The authors state that they used the 'most potent' shRNA of those present in the screen – but all three RhoU shRNAs tested were equally potent at reducing Rhou mRNA levels, thus the text needs altering here. Other RhoU, Trio and Fgd2 shRNAs replicate the initial hair follicle findings. This was done for two read-outs in Figure 4 only for RhoU, but not for other critical experiments that the authors spend most of the rest of the manuscript investigating i.e. effects on epidermal/keratinocyte cell shape, actin cytoskeleton and cell-cell junctions. These results need to be repeated with another RhoU shRNA to prove that they are not off-target effects.

We replaced “potent” with “validated” shRNA in our revised manuscript, as the reviewers are correct that there are multiple strong hairpins. We also agree that it is useful to confirm our findings with 2 independent *Rhou* shRNAs to verify that the observed defects are not off-target effects. In our initial manuscript, we showed that *shRhou-504* and *shRhou-505*-depleted skins have misoriented hair follicles (Figure 4A, B). Additionally, we showed that both *shRhou-504* and *shRhou-505*-transduced animals present an initial delay in epidermal barrier formation and a thinner epidermis at E18.5 (Figure 4—figure supplement 1B, D). Finally, we had shown (Figure 5—figure supplement 2A) that primary keratinocytes transfected with *shRhou-504* behave like those transfected with *shRhou-505* in exhibiting clear delays in adherens junction formation. Upon examining individual cells, we observed that transfection of both *shRhou-504* and *shRhou-505* resulted in an increase in F-ACTIN stress fibers and focal adhesion size (Figure 5—figure supplement 2B-D).

However, in response to the reviewers’ suggestion, we now also add a new Figure 5—figure supplement 1 to show that in vivo, actin dynamics, adherens junctions, levels of phosphorylated myosin II light chain and importantly cell shape are also perturbed with *shRhou-504,* analogous to what we show in Figure 5 for *shRhou-505.* Additionally, we now show in a new Figure 7—figure supplement 1Cthat similarly to the *shRhou505* hair pegs, the *shRhou-504* hair pegs are of comparable length to control hair pegs and we show in a new Figure 7—figure supplement 1D that *shRhou-504* placodes have a larger area than control placodes. Altogether, these new additions round off the essential need to document clearly that the defects we see in cell shape in vivo, as well as the defects in actin dynamics, adhesion and PCP,are not the results of off target effects.

2) Figure 5B: what is the variability between experimental results between mice? Is each cell an individual variable in the statistical analysis? The authors should rather consider each experiment (mouse) as an independent entity for comparison, because inevitably the knockdown will vary per mouse.

We thank the reviewers for raising this concern, although we did consider each cell to be an individual variable, it is important to point out that each transduced cell has a different integration site for the lentiviral DNA. Hence, the concern is not as great as for instance transgenic mice, and this is why we had analyzed the data on a cell by cell basis. Notably, we also always analyze embryos with comparable transduction efficiencies (≥ 80%) in order to limit variability between mice with regards to phenotypic analyses. We now have stated this information in the revised Materials and methods of our manuscript.

That said, following the reviewers’ suggestion, we now have gone back and modified our analysis and where needed for statistical significance, added another 1-2 embryos, so that we now consider each mouse as an independent entity. The new data are now presented in Figure 5B and Figure 5—figure supplement 1B where we use each mouse as an independent entity for comparison. Gratifyingly, both the results and the conclusions are similar to those obtained when we considered each cell as an independent entity. We agree that for the reader, this strategy will simplify the issue.

3) Figure 6: Presumably PAK2 rather than PAK1 was the strongest hit in the RhoU interactome because it is more highly expressed than PAK1 (or PAK3) in the epidermal cells used for the screen. This needs to be verified. To further support the possible involvement of PAK, activation of PAK1 and PAK2 should be tested in RhoU depleted keratinocytes by western blotting for phosphorylated forms of the kinases.

We thank the reviewers for raising this point. Indeed PAK2 is the more robustly expressed in the epidermal progenitors! As suggested, we’ve now analyzed the expression level of PAK proteins in primary epidermal keratinocyte lysates. To do so, we used a pan-PAK antibody that recognizes PAK1, PAK2 and PAK3. We now present a western blot with these data in a new Figure 6E and make this point in the text. Additionally, transcriptome profiling from primary keratinocytes (Fuchs lab RNA-seq data, unpublished data) revealed that expression of *Pak2* mRNA is more abundant than *Pak1* and *Pak3*. We also observed using transcriptome profiling from transduced E14.5 epidermal cells that *Pak2* mRNA is the most highly expressed *Pak* gene during epidermal development. We now present these data in Figure 6E.Together our mRNA and protein expression analyses are nicely complementary.

Finally, the reviewers requested that we examine the RHOU dependency of PAK1/PAK2 phosphorylation/activation in primary keratinocytes. We’ve now added western blotting for phosphorylated forms of the PAK kinases in a gain Figure 6F and loss of RHOU function Figure 6G setting to illuminate the importance of RHOU’s interactions with PAK kinases.

4) The cell shape changes as well as the changes in focal adhesions could all be explained by altered contractility, which has also been shown to be downstream of PAK. The authors could consider staining or western blotting for phosphorylated myosin light chain to strengthen the mechanistic aspect of the manuscript.

We thank the reviewers for raising this hypothesis and agree that this could help strengthen the mechanistic aspect of our manuscript. Following this suggestion, we have now stained *shScr, shRhou-504* and *shRhou-505* -transduced embryos with antibodies against the phosphorylated myosin II light chain (pMLC2) and we show that similarly to what is observed in the gut epithelium of *Rhou* knockout animal, depletion of RHOU increases the level of pMLC2 We now present this data in Figure 5C and Figure 5—figure supplement 1C. We now also shown that when we overexpress RHOU in vitro in primary keratinocytes, PAK1/2 become activated and this correlated with a decrease in pMLC2, conversely when we knock down RHOU, PAK1/2 show decreased activity and this correlates with increase in focal adhesions and actomyosin (Figure 5—figure supplement 1B-D). We now present this data in Figure 6F-G.

5) The screen identified Rac1 as a regulator of hair follicle morphogenesis, which is supported by hair formation defects in knockout mice. However, it also identified RhoB and Rac3, where corresponding knockout mice do not show obvious hair follicle defects. Cdc42 was surprisingly not detected as an important regulator of hair follicle formation, although it is known that Cdc42 knockout in keratinocytes has a significant effect on hair follicle maintenance. These discrepancies must be discussed in the context of the field.

We are grateful to the reviewers for raising this important point. As pointed out by the reviewers, we identified *Rhob* and *Rac3* as potent regulators of hair follicle morphogenesis, and yet both *Rhob* and *Rac3* knockout mice are viable and fertile without obvious defects during skin development (Liu et al., 2001 and Corbetta et al., 2005). We now explain in our Discussion that gene compensation mechanisms might be responsible for this discrepancy.

Intriguingly, new studies by Didier Stanier’s lab have unearthed some of the gene compensation machinery responsible for the mysterious differences in phenotypes previously observed between gene knockout and knockdown (El-Brolosy et al., 2019). Interestingly, they’ve demonstrated that these gene compensation mechanisms are triggered only in certain knockouts and not upon RNAi-mediated knockdown. This tantalizing recent discovery explains why shRNA knockdowns often reveal gene functions that were previously hidden through mechanisms of gene compensation in knockout animals. Although such probing is beyond the scope of our study here, we’ve now raised this issue in our Discussion. Indeed the particular case in point seems relevant, as it was previously observed that depletion of RHOA in the skin triggers an important upregulation of RHOB (Jackson et al., 2011). Importantly, potential gene compensation by other Rho GTPases was not addressed in *Rhob* knockout animals and this analysis was limited in the *Rac3* knockout, which could explain the lack of observed phenotypes in the knockouts but not the shRNA knockdowns.

Regarding the flip side of the coin, although the reviewers are correct in that *Cdc42* conditional knockout in the skin leads to defects in hair follicle maintenance, these defects are only manifested after birth. This suggests that the initial stages of hair follicle morphogenesis are not affected, most likely accounting for why *Cdc42* did not surface in our screen. We have now modified our Introduction to indicate that pups rather than embryos are affected by *Cdc42* loss and more clearly state the defects that those authors observed in their conditional knockout.